# Interpersonal physiological and psychological synchrony predict the social transmission of nocebo hyperalgesia between individuals

Rodela Mostafa[1], Nicolas Andrew McNair [1], Winston Tan [1], Cosette Saunders [1], Ben Colagiuri[1] & Kirsten Barnes [1,2] ✉

Witnessing another's pain can heighten pain in the observer. However, research has focused on the observer's *intra*personal experience. Here, a social transmission-chain explored the spread of socially-acquired nocebo hyperalgesia. Dyads of genuine participants were randomised to 'Generations' (G1–G3). G1-Demonstrators, observed by G2-Observers, experienced high/low thermal pain contingent on supposed activity/inactivity of a sham-treatment. G2 became Demonstrators, witnessed by G3-Observers. They experienced fixed low-temperature stimuli irrespective of sham-treatment 'activity'. G3 then Demonstrated for G4-Observers (a confederate), also experiencing low-temperature stimuli only. Pain ratings, electrodermal activity, and facial action units were measured. G1's treatment-related pain propagated throughout the chain. G2 and G3 participants showed heightened subjective and physiological response to sham-treatment, despite equivalent stimulus temperatures, and G3 never witnessing the initial pain-event. Dyadic *inter*personal physiological synchrony (electrodermal activity) and psychological synchrony (Observer's ability to predict the Demonstrator's pain), predicted subsequent socially-acquired pain. Implications relate to the interpersonal spread of maladaptive pain experiences.

Pain is a complex and near universal experience that involves psychological, cognitive, and social components[1–3]. Even though we often learn about pain through others, the influence of social factors remains understudied[4,5]. Understanding the impact that social interaction has on pain is important. Research concerning both human and non-human animals has demonstrated that exposure to stress in others, including those experiencing pain, plays an important role when appraising and navigating the environment[6,7]. This type of social exposure allows us to understand and empathise with the experiences of others[8] but can also signal the presence of noxious stimuli or other environmental risk factors[9]. However, while often adaptive, our propensity to learn from others can have unwanted consequences. The nocebo effect, a pervasive problem where negative expectancies amplify symptoms such as pain (termed nocebo hyperalgesia), concerns one such instance. While most research has focused on key mechanisms of nocebo hyperalgesia that are

non-social, growing evidence indicates that social learning plays a significant role in generating maladaptive pain experiences[10,11], with moderate to large effects on pain and expectations for pain in meta-analysis[12].

Observing another person experience pain from a treatment leads to heightened pain in the observer when they undergo the same treatment[13–15]. Importantly, this socially-acquired nocebo hyperalgesia can be observed in autonomic and brain activity[16–20] and even in response to visual and auditory stimuli associated with pain, but independent of a discrete medical treatment[16,18,21–31]. As such, nocebo hyperalgesia can be triggered in situations as diverse as receiving a vaccination[32] to participating in a sporting event[33]. Understanding the mechanisms of socially-acquired nocebo hyperalgesia is therefore critical for understanding the significant burden it causes. However, we still know little about how and when nocebo effects spread between individuals.

[1]School of Psychology, University of Sydney, Sydney, NSW, Australia. [2]School of Psychology, University of New South Wales, Sydney, NSW, Australia. ✉e-mail: kirsten.barnes@unsw.edu.au

Existing studies concerning socially-acquired nocebo hyperalgesia involve artificial social information, including fictitious pain ratings[16,17,31], photographic or video stimuli of social 'models'[13–15,18–20,22–27,30,34], or confederates feigning pain[21,28,35]. While pivotal for demonstrating that social learning can lead to nocebo hyperalgesia, these approaches do not capture the full complexity of social communication as it unfolds in the real world[36]. For example, photographic/video stimuli elicit differences in cognitive processing when compared to genuine in-person interaction[37–40], while observing someone pretend to experience pain is not equivalent to observing genuine pain, eliciting different behavioural and neural responses[41,42]. Most importantly, given the model's behaviour is predetermined, there is no opportunity for dynamic interpersonal interaction, which ignores the symbiotic nature of social communication[43], including interpersonal synchrony.

Interpersonal synchrony refers to shared, recursive experience between individuals and is thought to confer empathetic and prosocial benefits[44–46]. Research concerning social factors (e.g., patient-clinician alliance[47–49] and touch/handholding[50–52]) has shown that greater synchrony of facial expressions[49], as well as neural and physiological responses between dyads[47,48,50–52], can induce reductions in pain. However, synchrony may not be purely beneficial[53]. Model/observer synchrony in electrodermal activity (EDA) during fear-conditioning has been found to exacerbate autonomic arousal in the observer when directly exposed to the fear-conditioned stimulus[54,55]. The role of interpersonal synchrony in socially-acquired nocebo hyperalgesia has not been characterised. However, if synchrony amplifies social learning independent of whether the content is beneficial or harmful, then interpersonal synchrony could be a key mechanism of socially-acquired nocebo hyperalgesia that has evaded attention due to the current focus on artificial social models.

To test this, the present study investigated the spread of socially-acquired nocebo hyperalgesia along a transmission chain of genuine participant dyads. Participants rotated between demonstrating and observing pain experiences via a conditioning paradigm adapted to explore transmission of the nocebo effect. Participants were randomised to one of three 'generations' in the social transmission chain (G1/G2/G3) with a confederate taking the final position (G4). Participants assigned to G2 and G3 acted as Observers and then Demonstrators. Those assigned to G1 were only ever Demonstrators and acted as 'seeding' models who initiated the chain. The confederate (G4) only ever acted as an Observer and concluded the chain. G1 participants underwent standard conditioning, experiencing high/low-intensity painful stimuli contingent on the supposed activity/inactivity of a treatment (actually, a sham). G2 observed this, as per existing studies on socially-acquired nocebo-hyperalgesia. Critically, however, all other genuine participants (G2-G3) experienced low-intensity stimuli irrespective of treatment activity. When G2 experienced sham-treatment, G3 observed them, and when G3 underwent sham-treatment, G4 observed them, thereby creating a social transmission chain where subsequent generations could learn not only from observing direct nociceptive conditioning (G2), but actual nocebo hyperalgesia (G3), to the supposed treatment. As such, this paradigm allowed for genuine participants, rather than actors, to express pain to sham-treatment at all stages of the chain, but critically, to test whether interpersonal synchrony between dyads increased subsequent nocebo hyperalgesia in the observer. Further, due to the transmission chain containing multiple stages, whether a nocebo hyperalgesic response in one individual had the capacity to spread and cause a subsequent nocebo hyperalgesic response in another individual could be tested. Pain ratings, expectancies, pain-related facial expressions, and autonomic response were measured. Primary hypotheses were that socially-acquired nocebo hyperalgesia would propagate along the social transmission chain (from G2 to G3) and that greater interpersonal synchrony between dyads would be associated with stronger social transmission of nocebo hyperalgesia. All specific hypotheses are listed in the Methods section below.

## Methods

### Pre-registration
The design, hypotheses, and primary analysis plan were pre-registered on the 5th May 2022 prior to data collection (https://aspredicted.org/kd446.pdf), data and code are available via the Open Science Framework (https://osf.io/qzstu/?view_only=68a87230dcfe4245bb4dc5d0564cf0d9).

### Participants
One hundred and one healthy adults completed the study (70 identified as female, 24 as male, and 7 as non-binary; $M_{age}$ = 23.7 years). Sixty-nine were drawn from a community sample recruited via Facebook adverts and received AU$30 for their time. Thirty-two were students from the University of Sydney who participated via a psychology student participation scheme in exchange for course credit. All participants were pain-free at the time of testing, had normal or corrected-to-normal vision, were not taking any pain medication, self-reported as not being pregnant, and had not previously taken part in research concerning the effect of transcutaneous nerve stimulation (TENS) on pain. Participants provided informed written consent and were debriefed following the experiment. All experimental procedures were approved by the University of Sydney Human Research Ethics Committee (reference: 2022/287).

### Apparatus
The experiment was performed on two PCs running PsychoPy software[56]. Images were simultaneously displayed on two 27-inch LCD monitors (1920 × 1080-pixel resolution, 60 Hz refresh rate) at a viewing distance of ~90 cm. Communication and synchronisation between the two PCs was achieved via a TCP connection over the local area network.

**Thermal stimuli**. Thermal stimuli were delivered using a Pathway Model CHEPS device (Medoc, Ramat Yishai, Israel). The thermode was attached to the medial part of the participant's right volar forearm using a Velcro strap. In order to increase pain sensitivity[57,58] and avoid nociceptor fatigue[59] the baseline temperature was set to 40 °C, as in our previous research[60]. Two temperature destinations were employed: a 'Low' (45 °C) and 'High' (54 °C) temperature.

**Sham TENS device**. The sham treatment was described to participants as transcutaneous electrical nerve stimulation (TENS). This device consisted of a bar electrode connected to a stimulus isolator (Model FE180; ADInstruments, Bella Vista, NSW, Australia) and attached to the participant's right proximal volar forearm with surgical tape[61–64]. Participants were instructed that the TENS treatment would exacerbate their experience of pain, and this was reinforced with a short handout that participants read during the set-up procedure (see refs. 60,63). All participants were instructed that they would not be able to feel the activity of the TENS device once properly calibrated, thereby ensuring they would believe the device was active without any tactile stimulation that might interfere with measurements of the pain response. This was established through a pseudo-thresholding procedure, during which the current of the electrical stimulation (5 × 0.2 ms square pulses) was decreased from 2.7 mA in steps of 0.2–0.3 mA until no longer detectable (typically ~0.7 mA). No electrical stimulation was delivered during the experiment itself. Instead, supposed activation and deactivation of the TENS device was signalled by visual cues (green and blue squares; counterbalanced).

**Electrodermal activity (EDA)**. Electrodermal activity (EDA) was recorded from two stainless steel electrode plates attached to the distal phalanges on the index and middle fingers of the right hand. Data were recorded continuously via a PowerLab 4/26 DAQ device and Galvanic Skin Response bioamplifier (Model FE116; ADInstruments, Bella Vista, NSW, Australia), and digitized at a sampling rate of 1 kHz using LabChart software (Version 8, ADInstruments).

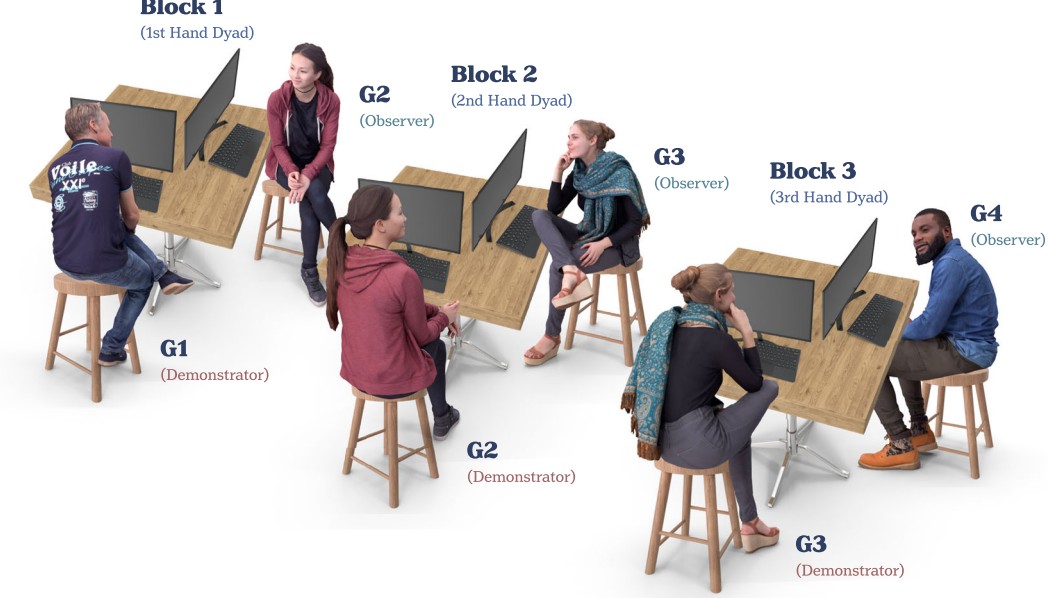

**Fig. 1 | Overview of the social chain (Blocks 1–3).** During Block 1 of the social transmission chain (containing the 1st Hand Dyad), the G1 (Demonstrator) undergoes the thermal procedure while the G2 (Observer; N = 36 biologically independent samples) watches. G1 leaves the room and G3 enters for Block 2 (2nd Hand Dyad), where G2 (now Demonstrator) undergoes the thermal procedure and G3 (Observer; N = 29 biologically independent samples) watches. G2 then leaves the room and G4 enters for Block 3 (3rd Hand Dyad). In this final round, G3 (now Demonstrator) undergoes the thermal procedure while G4 (Observer; a confederate) watches. Pre-registered experimental focus was on G2 and G3 participants as they had the opportunity to both receive and transmit social information. Image components under licence from Envato.

**Facial action units (FAU).** Facial expressions were recorded during the testing session via webcams (Logitech StreamCam; 1920 × 1080p, 60fps). Facial action units[65] relating to facial expressions of pain[66] were extracted using a facial behaviour analysis toolkit (OpenFace 2.2.0).

### Measures
**Demographic information.** Demographic information (age and self-report gender) of participants was collected via a Qualtrics form delivered to the participant's phone via a QR code.

**Expectancy ratings.** Both Demonstrators and Observers rated how painful they expected an upcoming thermal stimulus to be on a Visual Analogue Scale (VAS). This was presented in the centre of the monitor as a 100-point horizontal line. Two anchors were employed: 0 (not painful) and 100 (extremely painful). A changing number located above the scale indicated the current value underneath the mouse pointer.

**Pain ratings.** Following thermal stimulation, Demonstrators rated how painful it was on a VAS, presented in the centre of the monitor as a 100-point vertical line. Two anchors were employed: 0 (not painful) and 100 (extremely painful). A changing number to the right of the scale indicated the current value underneath the mouse pointer.

**TENS accuracy ratings.** At the end of each Testing Block, participants made a two-alternative forced-choice selection of the cue (green or blue square) that they thought was associated with activation of the TENS device. A confidence rating accompanied this choice. Confidence data was collected on a 7-point Likert-type scale (anchors: 0 = Not Confident; 7 = Confident).

**Manipulation check.** A manipulation check was presented at the end of the testing session. Participants were asked to describe the aims of the study (free response). No participants guessed the true aims (i.e., that equivalent temperatures were used to explore the nocebo effect).

### Design and procedure
A mixed design was employed, with a single between-subjects factor of Generation (G1–G3), and two within-subjects factors of Treatment Cue (Tx (TENS treatment) vs. NT (no treatment)) and Trial (1–6, per-Cue). For the between-subjects factor of Generation, participants were randomly allocated (via a random number generator) to one of three Generations: 1) Generation 1 (G1); 2) Generation 2 (G2); 3) Generation 3 (G3). A confederate occupied the last position, Generation 4 (G4). This was one of two authors (WT and CS) one male researcher and one female researcher, of roughly equivalent age to participants.

Each experimental session was divided into three separate Testing Blocks (although a minority of sessions had two; see below). During each Testing Block, a dyad of participants (1x Demonstrator and 1x Observer) took part in the thermal TENS procedure. As shown in Fig. 1, testing Blocks were ordered to form a social transmission chain as follows: Block1 (1st Hand Dyad: G1 as Demonstrator | G2 as Observer); Block2 (2nd Hand Dyad: G2 as Demonstrator | G3 as Observer); Block3 (3rd Hand Dyad: G3 as Demonstrator | G4 (confederate) as Observer). The purpose of G4 was simply to conclude the chain, ensuring that there were always two individuals—one Demonstrator and one Observer—in each Block. When not taking part in a Testing Block, the participants sat in a waiting room where they were separated by room dividers to prohibit conferral regarding the experimental procedures.

Participants were informed that the purported 'TENS' device may increase their pain sensitivity and that a green (or blue) square would indicate that the TENS was active while a blue (or green) square would indicate that the TENS was inactive. Colours were counterbalanced across, but not within, experimental sessions. Participants were explicitly told that the colour of the TENS Cue would be the same in all Testing Blocks during their session. Participants were additionally instructed that the study examined non-verbal communication regarding the effects of TENS on pain. They were informed that their objective was to learn which Cue (blue or green) indicated that the TENS device was active or inactive. This would be achieved either through acting as a Demonstrator, who directly experienced the TENS treatment and

**Fig. 2 | Trial structure.** Single trial structure presented simultaneously to Demonstrators and Observers within the same dyad.

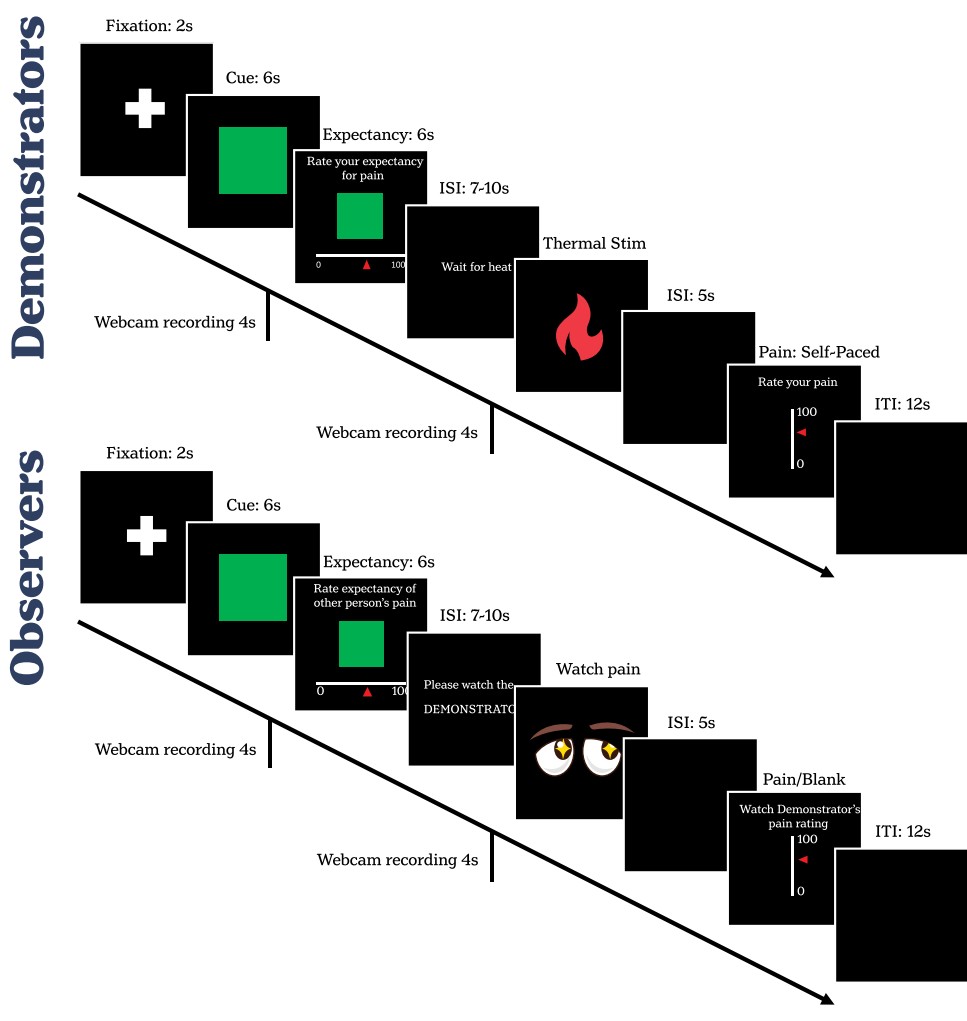

thermal stimuli, or as an Observer who watched the Demonstrator convey their experience non-verbally. Importantly, the experimenter was blind to which Cue was associated with TENS activity and therefore could not communicate this information to participants. Further, all participants were familiarised with the thermal stimuli prior to testing. During this process they were told that the efficacy of the TENS varied across individuals and that they would be shown a range of temperatures, from low (42 °C) to high (54 °C), demonstrating how the TENS *might* feel. As such, any expectancies regarding *when* the TENS was active and *how much* it modulated pain were learnt through social observation. Participants were not alerted to the fact that the study concerned nocebo hyperalgesia. The confederate (G4 as Observer) was also kept blind to which Cue had been associated with the sham-treatment.

Generations 1–3 played different roles in the social transmission chain. The purpose of G1 was simply to initiate the chain. As such, G1 participants only ever acted as Demonstrator. As the TENS device was a sham-treatment, it was made to appear effective in this group by surreptitiously varying the intensity of the thermal stimulus so that the High temperature (54 °C) was consistently paired with the Tx Cue (TENS active), and the Low temperature (45 °C) with the NT Cue (TENS inactive). Discrepancy in pain expression to the TENS therefore acted as a 'treatment event' (with natural variation across sessions) through which observational learning could take place.

In contrast, when G2 and G3 participants acted as Demonstrator, thermal stimuli were always delivered at the Low temperature, irrespective of the Cue. As such, any discrepancy in physiological or behavioural responses to Tx vs. NT among these participants was indicative of socially-

acquired nocebo hyperalgesia. The expression of socially-acquired nocebo hyperalgesia among G2-Demonstrators was generated by observing responses in G1-Demonstrators that were triggered by genuine differences in nociceptive input (similar to previous social learning research, but with genuine naïve Demonstrators). However, the expression of hyperalgesia among G3-Demonstrators was socially acquired from a response to treatment among G2-Demonstrators that was generated purely via expectancies (i.e., a nocebo effect generating a subsequent nocebo effect). Finally, G4 participants only ever acted as the Observer. Despite being a confederate, they followed an identical procedure to all other Observers. Their physiological response and self-report ratings were recorded to uphold the cover story but were not analysed. To ensure that they followed an identical procedure to the genuine participants, they were also asked to determine which of the two cues was associated with TENS activity in that session.

The trial structure is summarised in Fig. 2 and differed dependent on when participants were assigned to Demonstrator or Observer roles. During each Testing Block, Demonstrators and Observers sat facing each other on two sides of a large desk. Their respective monitors were positioned at such an angle that each could easily see their own screen and the face of the other participant. However, neither could see the screen of the other individual. For both participants, each trial began with a fixation cross displayed centre screen for 2 s. They were then both presented with either the same green or blue square for 6 s. Subsequently, Demonstrators and Observers were independently given 6 s to complete their expectancy rating for the upcoming thermal stimulation using a mouse with their left hand to click on the onscreen VAS. This was followed by a variable ITI of 7–10 s, during which time the phrase 'Wait for heat' was presented on the Demonstrator's

monitor and 'Please watch the DEMONSTRATOR' on the Observer's monitor. The thermal stimulus was then applied to the Demonstrator's arm, accompanied by an auditory tone alerting the Observer to the fact that the event had occurred. A blank screen was presented for 5 s after the onset of the heat stimulus. Demonstrators were then required to rate their pain on the onscreen VAS in a self-paced manner. On 50% of trials, the changing pain ratings as the Demonstrator moved their mouse along the VAS, as well as their final ratings, were displayed in real-time on the Observer's monitor. On the remaining trials, a blank screen was shown instead. To ensure relative consistency across dyads, the pain ratings from the first Tx and the first NT stimulus were always shown. Overall, three pain ratings were shown from the first half of the block and 3 from the last, while no more than two pain ratings were shown in succession. Half of the pain ratings were suppressed to ensure that Observers paid attention to the facial expressions of the Demonstrator and could not rely solely on pain ratings to receive social information, which could have reduced synchrony. Finally, a blank ITI was presented for 12 s before the next trial began. The webcams recorded the facial expressions of both participants at two time periods: (1) $FAU_{cue}$ (indicative of anticipatory responses)—this epoch occurred 1 s before the presentation of the Cue to 3 s after; (2) $FAU_{heat}$ (indicative of phasic pain responses)—this epoch occurred 1 s prior to delivery of the thermal stimulus to 3 s after).

Each Testing Block consisted of the presentation of 6 Tx, and 6 NT, Cues. To avoid serial order carry over effects, trial order was pseudorandomised so that each trial type followed every other trial type with equal probability relative to their occurrence[67]. Additionally, it was ensured that there were no more than 3 repeated trials with the same Cue in a row, and that each Cue was shown no less than 2 and no greater than 4 times in each half of the 12-trial Testing Block. Three sequences adhering to these rules were created, each beginning with a NT Cue. These sequences were then inverted, to produce another three sequences starting with a Tx Cue. Presentation of these blocks were randomised across sessions using a random number generator.

Participants were tested in a single 1.5 h session. Ideally, each session comprised three participants ($N_{sessions} = 29$; $N_{participants} = 87$) plus the confederate. However, so as not to waste time and resources, sessions were still run if one participant did not arrive ($N_{sessions} = 7$; $N_{participants} = 14$). This meant that there were slightly fewer G3 ($N = 29$) participants than G2 ($N = 36$).

### Data pre-processing and statistical analysis

**Pre-processing of phasic data.** Pre-processing of phasic EDA data was conducted in MATLAB (v2022a; Mathworks Inc.), while webcam data was processed with Openface (2.2.0) to produce FAU metrics.

**EDA data.** The EDA data were band-pass filtered (0.0159–5 Hz) before being divided into two epochs: (1) $EDA_{cue}$ (2 s pre-cue–6 s post-cue) corresponding to anticipatory arousal; and (2) $EDA_{heat}$ (2 s pre-heat–6 s post-heat) corresponding to phasic arousal. The data were then downsampled to 100 Hz. Consistent with our previous research[60], trials with any datapoints with an amplitude more than 4.89 SDs from the average, computed separately for $EDA_{cue}$ and $EDA_{heat}$ epochs, were excluded from further analysis (i.e., extreme outliers: average 0.24 trials per participant). A jack-knifing procedure was used to calculate the SDs, to reduce the effect of outliers on the exclusion criterion. Peak-to-peak $EDA_{cue}$ and $EDA_{heat}$ responses were calculated on each trial as the maximum amplitude from 3 s post cue/heat onset to the end of the epoch, minus the minimum amplitude from that point back to the cue/heat onset. Values lower than 0.02μS were assigned a value of 0. These data were log-transformed, after adding 1 to every value.

**FAU data.** Webcam data, recorded during the cue and heat epochs (see above), were processed with OpenFace software to extract FAUs specifically associated with pain expressions [AU: 04, max(06, 07), max(09, 10), 12, max(25, 26)[66]]. The values were summed to produce a 'pain

expression' score that indicated the strength with which pain-related facial action-units were engaged at each timepoint. The maximum value in the 3 s following cue or heat onset was used as a measure of pain-related expression in each epoch of each trial.

**Physiological synchrony (EDA).** Physiological timeseries data were analysed with cross recurrence quantification analysis (CRQA), using the *crqa* package in R[68]. This method quantifies recurrence in the data of two individuals[69]. Here, we explored the extent to which the temporal organisation of the Demonstrator and Observer's autonomic response adapted, coupled, and integrated over time. CRQA achieves this via cross recurrence plots (CRPS) which represent the co-visitation patterns of two datasets - how one timeseries revisits states that the other had previously visited.

Raw EDA data was extracted for both the Demonstrator and Observer across the entirety of the Testing Blocks that involved the following dyads: (1) Block1_Synchrony (G1-Demonstrator with G2-Observer); (2) Block2_Synchrony (G2-Demonstrator with G3-Observer). Following previous research[54], data from each dyad were band-pass filtered (0.05–1 Hz), downsampled to 8 Hz, and then z-scored, with the optimal parameters (radius, delay, and embedding dimensions) required for the CRQA estimated using a model-fitting procedure parameters that yielded an average recurrence rate of 2–4%.

We extracted two CRQA parameters: Determinism (DET) and Laminarity (LAM); previously shown to predict socially-acquired fear learning in EDA data[54]. An exponential transformation was applied to both variables prior to obtaining z-scores to correct for left skew. Increasing values of DET are associated with a greater number of recurrent timepoints forming a diagonal line in the CRPS (the rate of recurrence or co-visitation). This can be interpreted as an increased connection in trajectories over time (i.e., interconnection). Greater LAM values are associated with increasing points on the vertical line of the CRPS (the rate of covariation between the two responses). This suggests greater time periods of stability in the autonomic response of both individuals (i.e., smooth periods in the mutual timeseries).

**Psychological synchrony (expectancy ratings).** Correlation in pain Expectancy between the Demonstrator and Observer's ratings (Block1_Synchrony and Block2_Synchrony) were explored as a metric of psychological synchrony. Specifically, we tested whether those Observers who were better at predicting how much pain the Demonstrator thought they would experience on each trial (i.e., were more in synch with their expected experience) would go on to demonstrate greater nocebo hyperalgesia. As previous research has demonstrated that expectancies feed into the perception of pain at the level of each trial[70], we chose to focus on Demonstrator expectancies, rather than pain ratings, as we wanted to measure synchrony for the same cognitive process – belief about the upcoming thermal stimulus – occurring at the same point in time for both participants.

We note that this measure of 'Psychological Synchrony' differs from physiological measures of synchrony as the expectancy metric was only continuous at the level of the trial. Instead, in the present study, we use the terms physiological and psychological synchrony to broadly refer to intercorrelations between participants at the level of the individual dyads. As timeseries data at the trial level was not available (i.e., there was only one expectancy measure per trial), correlation coefficients were calculated for each dyad to represent the extent of synchronous responding. This metric therefore focused on the raw rate of occurrence (cross-correlation), without the metric of cross-recurrence that CRQA additionally provides[68].

**Hypotheses.** Hypotheses were: $H_1$) G1-Demonstrators (who experienced differences in nociceptive intensity) would demonstrate elevated pain ratings, EDA, and pain-related FAUs to supposed sham-treatment 'activity' (manipulation check); $H_2$) G2- and G3-Demonstrators would demonstrate nocebo hyperalgesia (subjective ratings, EDA, and FAU),

despite never experiencing the high-intensity stimulus (primary analysis); $H_3$) $H_2$ effects would be moderated by chain position (primary analysis: G2 > G3). We further hypothesised that: $H_4$) socially-acquired nocebo hyperalgesia (in G2- and G3-Demonstrators) would be positively associated with enhanced prior synchrony between G1-Demonstrator/G2-Observer and G2-Demonstrator/G3-Observer dyads; and, $H_5$) the effect of this synchrony on nocebo hyperalgesia among G3-Demonstrators would be reduced if G2-Demonstrators were categorised as nocebo non-responders (manipulation check).

### Statistical analysis

**Manipulation check (G1-Demonstrators only).** To ensure that the High and Low thermal stimuli were sufficient to generate differences in the G1 pain response to Tx and NT cues, all phasic outcomes (pain ratings, expectancy ratings, $FAU_{cue/heat}$ and $EDA_{cue/heat}$) were analysed via within-subjects ANCOVAs with Cue and Trial as factors and Gender as the covariate (see: $H_1$). For all analyses, an alpha of .05 was employed and two-tailed tests are reported. Data distributions were assumed to be normal, but these were not formally tested. All analyses were conducted using the R software package (version 4.3.1)[71]. ANCOVAs were run with the afex package[72] in conjunction with the emmeans package[73]. 95% CIs around effect sizes are calculated with the effectsize package and are reported using values for a 2-sided test[74].

**Manipulation check (responders vs. non-responders).** It was expected that some participants would be non-responsive to the nocebo manipulation, as has previously been observed in behavioural and neuroscientific research e.g., refs. [20,75]. These participants would be expected to curtail the transmission of social information regarding nocebo hyperalgesia[76], instead communicating that the treatment was ineffective. $H_5$ tested for a reduced effect of synchrony on pain-modulation among G3-Demonstrators if G2-Demonstrators were classified as a non-responder. To disentangle these types of social information, we performed a Monte Carlo permutation analysis (10,000 random permutations) on each subject's pain rating data. An observed nocebo response (average Tx minus NT pains ratings) in the upper 95% of the estimated distribution classified people as a Responder. All others were classified as Non-Responders. Whether these two groups differed in their physiological response and facial expressions is reported to demonstrate responsivity/non-responsivity across all outcomes. Distributions of the raw data, including within-subject differences between Tx and NT Cues, for G1 Demonstrators, can be found in Supplementary Fig. 1.

**Primary analysis (social transmission of pain).** To test $H_2$-$H_3$, a mixed measures ANCOVA was run, with a single between-subjects factor of Generation (G2, G3), and within-subjects factors of Cue (Tx vs. NT) and Trial (1–6). Note that the Cue factor was erroneously omitted in our pre-registration when describing this analysis, but provides the critical test of nocebo hyperalgesia between generations, including the magnitude of the pain ratings, and therefore was the intended analysis. Given its strength in moderating socially transmitted nocebo effects, whether the gender of the dyad members was the same (matched vs. unmatched) was included in the analysis as a covariate[28,77]. Trial number was included in the analysis to assess extinction of the nocebo effect. Pain Ratings formed the primary outcome. Distributions of the raw data, including within-subject differences between Tx and NT Cues, for G2 and G3 Demonstrators across the primary and secondary outcomes can be found in Supplementary Fig. 2.

**Secondary analysis (Phasic EDA and FAU Data).** Phasic measures of $EDA_{cue}$, $EDA_{heat}$, $FAU_{cue}$ and $FAU_{heat}$ were analysed using the same ANCOVA model as outlined above (testing $H_2$-$H_3$). As missing data existed (e.g., due to removal of outliers; 60 trials total—average 0.3 trials per participant), to reduce listwise deletion sequential pairs of Trials (Tx and NT) were averaged together (resulting in a within-subject factor of Trial with three levels: 1 (average of trials 1–2); 2 (average of trials 3–4); 3

(average of trials 5–6)). This merging meant that only a single trial-pair was missing across all participants.

**Secondary analysis (effect of synchrony on nocebo hyperalgesia).** To test $H_4$, hierarchical regression models were run to test the effect of synchrony (psychological and physiological) among G1-Demonstrator/G2-Observer and G2-Demonstrator/G3-Observer dyads, on the magnitude of nocebo hyperalgesia among G2 and G3 participants respectively, when they subsequently became Demonstrators. Synchrony metrics (z-scores) were added in stage 1 and Generation (factorised) in stage 2. These stages therefore represented the main effect of Synchrony on the nocebo effect (stage 1), when controlling for Generation (stage 2). The final stage contained the Synchrony x Generation interaction (with the individual predictors representing conditional effects). Our original pre-registered analysis to test $H_5$ involved adding the dichotomous responder status of G2 to the hierarchical regressions outlined above, with the prediction that the effect of synchrony on pain-modulation at G3 would be reduced if G2 was categorised as a non-responder. However, this analysis did not allow for an exploration of the effect of the magnitude of the pain difference (Tx – NT) among G1 participants interacting with synchrony to predict nocebo hyperalgesia in G2 as all G1 participants were found to be responders in the present study. As such, we modified this analysis to run two regressions, one with G2 and one with G3 nocebo hyperalgesia as the outcome and used the pain difference (Tx – NT) of G1 and G2-Demonstrators as the predictor interacting with dyadic synchrony. This meant that nocebo hyperalgesia in both Generations could be compared using the same linear predictor (i.e., pain difference of the previous Demonstrator). However, the pattern of results was identical for G3 participants when employing the categorical responder/non-responder variable.

As some participants may not have entered their expectancy ratings within the time limit (6 s), there were some trials where expectancy ratings were missing from the dyad. This affected 100 trials (8.2%) in total. Two participants (both Observers) failed to select their expectancy ratings on either all or all-but-one trial, and those dyads were excluded from the synchrony analysis. Of the remaining data, on average 0.77 trials were missing from each dyad.

### Reporting summary

Further information on research design is available in the Nature Portfolio Reporting Summary linked to this article.

## Results

### Manipulation check 1: generation 1 demonstrators

It was hypothesised that G1-Demonstrators (who experienced differences in nociceptive intensity) would demonstrate elevated pain ratings, EDA, and pain-related FAUs to supposed sham-treatment 'activity' ($H_1$: manipulation check). An effect of Cue (Tx > NT) was found for all outcomes: Pain ratings ($F(1, 34) = 215.93$, $p < 0.001$, $\eta_p^2 = 0.86$, 95% CI[0.77, 0.91]); Expectancy ratings ($F(1, 32) = 56.08$, $p < 0.001$, $\eta_p^2 = 0.64$, 95% CI[0.42, 0.76]); $FAU_{Heat}$ ($F(1, 34) = 33.63$, $p < 0.001$, $\eta_p^2 = 0.50$, 95% CI[0.25, 0.66]); $EDA_{Heat}$ ($F(1, 34) = 101.61$, $p < 0.001$, $\eta_p^2 = 0.75$, 95% CI[0.59, 0.84]); $EDA_{cue}$ ($F(1, 34) = 6.55$, $p = 0.015$, $\eta_p^2 = 0.16$, 95% CI[0.01, 0.38]). This was with the exception of $FAU_{cue}$, which did not reach statistical significance ($F(1, 34) = 3.16$, $p = 0.085$, $\eta_p^2 = 0.09$, 95% CI[0.00, 0.29]). On average, therefore, G1 participants appeared sufficiently conditioned to transmit pain-related information. Additional statistical information can be found in Supplementary Note 1 and Supplementary Fig. 3.

### Primary analysis: pain ratings

It was hypothesised that G2- and G3-Demonstrators would demonstrate nocebo hyperalgesia (Tx > NT in subjective ratings, EDA, and FAU), despite never experiencing the high-intensity thermal stimulus ($H_2$), and that the strength of nocebo hyperalgesia would be moderated by chain position (G2 > G3: $H_3$). We therefore tested whether the expression of pain-

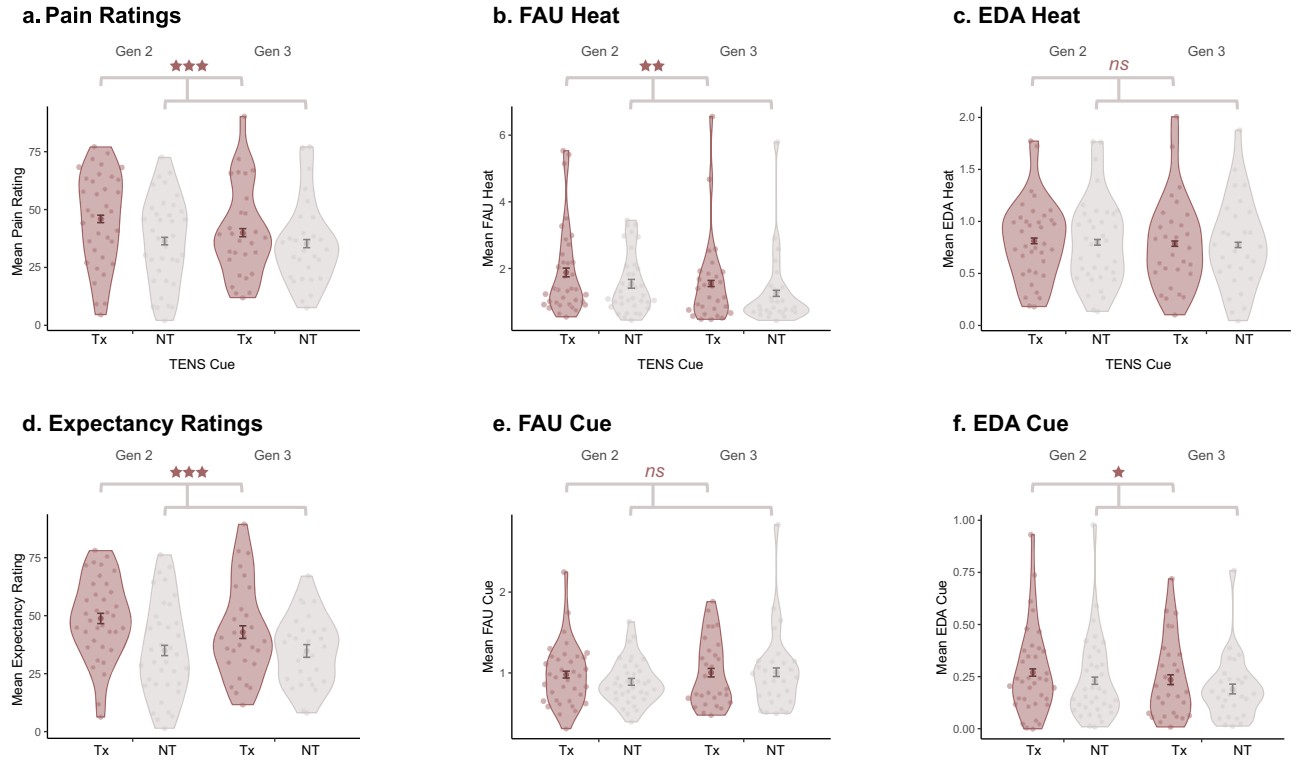

**Fig. 3 | Treatment vs. no treatment responses for G2 and G3.** Mean responses across all trials for the following outcomes: (**a**) Pain Ratings; (**b**) FAU Heat; (**c**) EDA Heat; (**d**) Expectancy Ratings; (**e**) FAU Cue; (**f**) EDA Cue. Responses are separated by Generation for readers interested in viewing the mean differences by experimental condition. However, we note that the main effect of Generation, as well as the Generation by Cue interaction, did not reach a level of statistical significance for all outcomes. Brackets with significance values therefore relate to the main effect of Cue collapsing across Generation (i.e., significance stars represent significant Tx

('Treatment') vs. NT ('No Treatment') comparisons only). Among G2 ($N = 36$ biologically independent samples) and G3 Demonstrators ($N = 29$ biologically independent samples), both the Tx trials were paired with the Low Temperature Destination (45 °C) with responses indicative of nocebo hyperalgesia. Error bars represent $+/-$ 1SED, adjusted for the within-subjects comparison (using the *afex* package in R[72]). Significance levels are depicted as (***$p < .001$); (**$p < 0.01$), and (*$p < 0.05$).

related information drove the transmission of nocebo hyperalgesia from G2-to-G3, across the entirety of the sample (i.e., collapsing across Responder-Status, see below). As depicted in Fig. 3, a statistically significant main effect of Cue on Pain Ratings was observed among G2 and G3 participants ($F(1, 62) = 18.14$, $p < 0.001$, $\eta_p^2 = 0.23$, 95% CI[0.07, 0.39]). Greater pain was experienced to Tx ($M_{adj} = 42.99$, $SE = 2.48$, 95% CI[38.04, 47.93]) than to NT ($M_{adj} = 35.84$, $SE = 2.31$, 95% CI[31.21, 40.46]), indicative of nocebo hyperalgesia. A statistically significant main effect of Trial was also present ($F(5, 310) = 12.35$, $p < .001$, $\eta_p^2 = 0.17$, 95% CI[0.09, 0.23]) with sensitisation occurring over time, irrespective of Cue. However, while nocebo hyperalgesia was numerically reduced at G3 (see Fig. 3), there was no statistically significant main effect of Generation ($F(1, 62) = 0.62$, $p = 0.435$, $\eta_p^2 = 0.01$, 95% CI[0.00, 0.11]) nor a Generation by Cue interaction ($F(1, 62) = 2.08$, $p = 0.154$, $\eta_p^2 = 0.03$, 95% CI[0.00, 0.04]). The Tx vs. NT difference therefore did not differ to a statistically significant level between Generations, providing support for H2 but a lack of support for H3.

**Secondary analyses: anticipatory and phasic responses**
The effect of social modelling on secondary outcomes (FAUs, EDA, and Expectancy Ratings) are presented in Fig. 3b–f. Additional statistical analyses are outlined in Supplementary Note 2. In summary, statistically significant main effects of Cue were observed for EDA$_{cue}$ ($F(1, 62) = 4.31$, $p = 0.042$, $\eta_p^2 = 0.07$, 95% CI[0.00, 0.21]), Expectancy ratings ($F(1, 62) = 19.96$, $p < 0.001$, $\eta_p^2 = 0.24$, 95% CI[0.08, 0.41]), and FAU$_{heat}$ ($F(1, 62) = 7.28$, $p = 0.009$, $\eta_p^2 = 0.11$, 95% CI[0.01, 0.26]). There was no statistically significant effect of Cue on FAU$_{cue}$ ($F(1, 62) = 0.71$, $p = 0.403$, $\eta_p^2 = 0.01$, 95% CI[0.00, 0.11]), and EDA$_{heat}$ ($F(1, 62) = 0.27$, $p = 0.605$,

$\eta_p^2 < 0.01$, 95% CI[0.00, 0.09]). The former is arguably unsurprising given this outcome did not differ by Cue among G1 participants to a statistically significant level. The latter may be explained by Non-Responders (as outlined in Supplementary Note 3). There were no statistically significant interactions between Cue and Generation, except for FAU$_{heat}$ which obtained threshold statistical significance. As such, the effect of social information on nocebo hyperalgesia was expressed in both self-report and physiological outcomes and appeared to be transmitted down the entirety of the chain.

**Interpersonal psychological synchrony**
It was hypothesised that socially-acquired nocebo hyperalgesia (Pain Difference in G2- and G3-Demonstrators) would be positively associated with enhanced prior synchrony between G1-Demonstrator/G2-Observer and G2-Demonstrator/G3-Observer dyads (H4). Results from regression models concerning psychological (Expectancy) and physiological (EDA) synchrony are presented in Table 1. The pain difference (Tx – NT trials) for all generations is presented in Fig. 4a. Due to missing Expectancy ratings, psychological synchrony ratings could not be calculated for two dyads (1x Gen1/Gen2 dyad and 1x Gen2/Gen3 dyad). These participants were therefore removed from both the psychological and physiological synchrony analyses so that an identical sample of participants ($N = 63$) were compared across all synchrony metrics.

The average standardised coefficient representing the strength of the association between Observer and Demonstrator expectancies had a medium effect size (Descriptive Statistics: $M = 0.34$, $SD = 0.45$), suggesting generally high levels of psychological synchrony between participants.

**Table 1 | Hierarchical regression output**

| | Model Summary / ANOVA | | | Coefficients | | | | | |
|---|---|---|---|---|---|---|---|---|---|
| | $R^2_{adj}$ | F | p | B | SE | β | t | p | 95% CI |
| Psychological Synchrony: Expectancy Ratings | | | | | | | | | |
| Step 1 | 0.04 | 3.71 | 0.059 | | | | | | |
| Synchrony | | | | 3.09 | 1.71 | 0.23 | 1.81 | 0.075 | [−0.33, 6.50] |
| Step 2 | 0.05 | 1.88 | 0.175 | | | | | | |
| Synchrony | | | | 3.09 | 1.70 | 0.23 | 1.82 | 0.074 | [−0.31, 6.48] |
| Generation | | | | 2.18 | 1.68 | 0.16 | 1.30 | 0.200 | [−1.18, 5.54] |
| Step 3 | 0.15 | 8.32 | **0.005** | | | | | | |
| Synchrony | | | | 3.62 | 1.61 | 0.26 | 2.25 | **0.029** | [0.40, 6.85] |
| Generation | | | | 2.18 | 1.59 | 0.16 | 1.37 | 0.175 | [−1.00, 5.35] |
| Synchrony:Generation | | | | −4.65 | 1.61 | −0.34 | 2.88 | **0.005** | [−7.88, −1.42] |
| Physiological Synchrony: Determinism (DET - recurrence in the EDA response) | | | | | | | | | |
| Step 1 | 0.07 | 5.73 | **0.020** | | | | | | |
| Synchrony | | | | 3.93 | 1.66 | 0.29 | 2.36 | **0.021** | [0.61, 7.26] |
| Step 2 | 0.09 | 2.31 | 0.134 | | | | | | |
| Synchrony | | | | 4.13 | 1.65 | 0.30 | 2.50 | **0.015** | [0.83, 7.43] |
| Generation | | | | 2.50 | 1.65 | 0.18 | 1.52 | 0.135 | [−0.80, 5.79] |
| Step 3 | 0.09 | 1.31 | 0.258 | | | | | | |
| Synchrony | | | | 3.51 | 1.73 | 0.26 | 2.03 | **0.047** | [0.04, 6.98] |
| Generation | | | | 2.47 | 1.64 | 0.18 | 1.50 | 0.138 | [−0.82, 5.76] |
| Synchrony:Generation | | | | 1.98 | 1.73 | 0.15 | 1.14 | 0.258 | [−1.49, 5.45] |
| Physiological Synchrony: Laminarity (LAM - stability in the EDA response) | | | | | | | | | |
| Step 1 | 0.01 | 1.95 | 0.168 | | | | | | |
| Synchrony | | | | 2.33 | 1.70 | 0.17 | 1.38 | 0.173 | [−1.05, 5.73] |
| Step 2 | 0.03 | 2.11 | 0.152 | | | | | | |
| Synchrony | | | | 2.61 | 1.69 | 0.19 | 1.55 | 0.128 | [0.19, 0.18] |
| Generation | | | | 2.47 | 1.70 | 0.18 | 1.45 | 0.152 | [−0.94, 5.88] |
| Step 3 | 0.04 | 1.22 | 0.275 | | | | | | |
| Synchrony | | | | 1.74 | 1.86 | 0.13 | 0.94 | 0.352 | [−1.98, 5.47] |
| Generation | | | | 2.38 | 1.70 | 0.17 | 1.40 | 0.168 | [−1.03, 5.78] |
| Synchrony:Generation | | | | 2.05 | 1.86 | 0.15 | 1.10 | 0.274 | [−1.67, 5.78] |

Regression output for Demonstrator/Observer Synchrony (psychological and physiological) on the magnitude of nocebo hyperalgesia when the Observer subsequently became the Demonstrator. For Psychological and Physiological Synchrony (Generation 2 $N = 35$ biologically independent samples; Generation 3 $N = 28$ biologically independent samples, due to missing data). Significant $p$-values are in bold.

Psychological synchrony (reflecting the Observer's ability to predict how much pain the Demonstrator thought they would experience) was found to interact with Generation (G2 vs. G3) to predict the magnitude of nocebo hyperalgesia to a statistically significant level (Tx–NT difference score; $B = −4.65$, $t(59) = 2.88$, $p = 0.005$, 95% CI[−7.88, −1.42]). The interaction is plotted in Fig. 4b. Simple slopes revealed that the slope for G2 was not statistically significant ($B = −1.0$ 95% CI[−5.32, 3.26]), with a difference in Tx/NT pain ratings of ~10 VAS points being reported across differing levels of synchrony. However, the slope for G3 was statistically significant ($B = 8.27$ 95% CI[3.45, 13.09]). The greatest difference between Generations occurred at low synchrony, where a reduction in the ability of G3 Observers to predict the pain expected by G2 Demonstrators reduced nocebo hyperalgesia when these Observers subsequently became Demonstrators. Specifically, synchrony in the ability to predict upcoming pain appeared to increase the nocebo effect among those watching a nocebo hyperalgesic response themselves.

**Interpersonal physiological synchrony: DET**
Recurrence or co-visitation (DET) in the EDA time course data between Demonstrator/Observer dyads positively predicted the magnitude of nocebo hyperalgesia experienced by the Observer when they subsequently became the Demonstrator ($B = 3.93$, $t(61) = 2.36$, $p = 0.021$, 95% CI[0.61, 7.26]) and remained statistically significant when controlling for Generation ($B = 4.13$, $t(60) = 2.50$, $p = 0.015$, 95% CI[0.83, 7.43]). There was no statistically significant interaction between DET and Generation ($B = 1.98$, $t(59) = 1.14$, $p = 0.258$, 95% CI[−1.49, 5.45]). Specifically, synchrony in the Demonstrator/Observer autonomic response predicted the magnitude of the nocebo hyperalgesic response in a separate experimental block when the original Demonstrator was no longer present. One difference between physiological and psychological measures of synchrony (as presented above) is that the physiological metrics are calculated across the entire time course of the experimental block and are therefore not tied in a phasic manner to the presentation of the thermal stimuli, whereas the Expectancy ratings are specific to the upcoming stimulus on each trial.

**Interpersonal physiological synchrony: LAM**
Smooth periods, or stability, in the mutual EDA timeseries (LAM) was not found to predict nocebo hyperalgesia or interact with Generation to a statistically significant level (see Table 1).

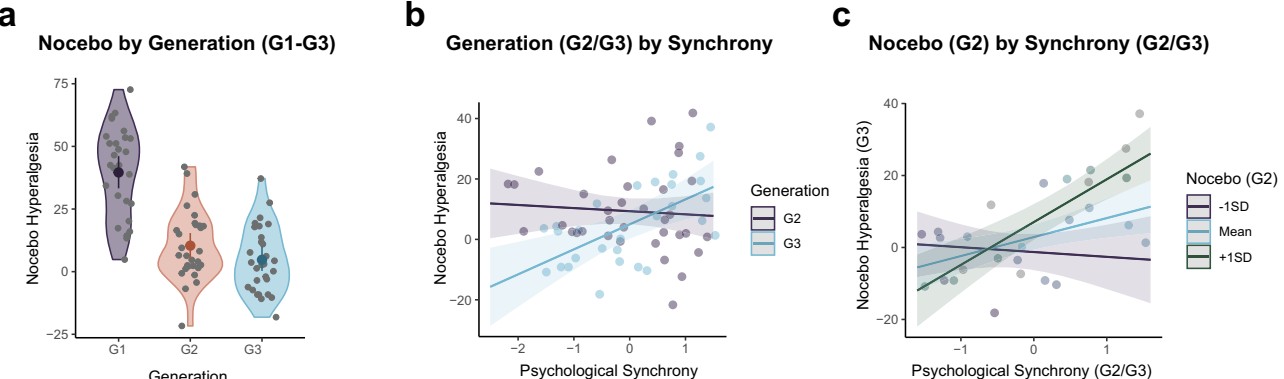

**Fig. 4 | Mean pain differences between groups and interactions between Generation and Psychological Synchrony.** (**a**) Mean difference in pain ratings between TENS Treatment (Tx) and No Treatment (NT) trials for Generation 1 ($N = 36$ biologically independent samples), Generation 2 ($N = 36$ biologically independent samples), and Generation 3 ($N = 29$ biologically independent samples). Error bars are $+/- 1\,SD$, not $SED$ (please note that while the y-axis is labelled 'nocebo hyperalgesia', G1 represents a pain response to direct nociception); (**b**) Interaction between Psychological Synchrony and Generation (G2/G3) on nocebo hyperalgesia (G2 $N = 35$ and G3 $N = 28$ biologically independent samples due to one missing datapoint per generation); (**c**) Interaction between G2 nocebo hyperalgesia (represented as slopes at $+/-1SD$ and the mean) and Psychological Synchrony (G2/G3; x-axis) on G3 nocebo hyperalgesia (y-axis). Analysis consists of $N = 28$ dyads (the total number present at G3 accounting for missing data).

## Interaction between interpersonal synchrony and the pain response expressed by the previous demonstrator

It was hypothesised that the effect of synchrony on the magnitude of subsequent nocebo hyperalgesia would be reduced in instances where the Demonstrator was less responsive to the treatment manipulation (H₅).

Separate hierarchical regressions were run with magnitude of G2 and G3's nocebo hyperalgesia (Tx – NT pain difference) when acting as Demonstrator as the outcome. This was run on data where a full chain of participants existed (G1-G2-G3), meaning that the same G2 participants were included in both analyses. Because there were two cases where psychological synchrony could not be calculated, the total number of G1/G2 and G2/G3 dyads was $N = 28$. The pain difference score of the previous Demonstrator (G1/G2, respectively) was added to the model in the first step, synchrony between dyads was added in the second, and the interaction between both predictors in the third. Note that pain difference is driven by genuine nociceptive differences at G1, but not G2.

In the case of psychological synchrony, the pain difference (Tx - NT) expressed by G1-Demonstrators did not predict the magnitude of nocebo hyperalgesia expressed by G2-Demonstrators to a level of statistical significance ($B = 0.14$, $t(25) = 0.91$, $p = 0.370$, 95% CI[−0.17, 0.45]) even when psychological synchrony was controlled ($B = 0.15$, $t(24) = 0.95$, $p = .359$, 95% CI[−0.17, 0.47]. Psychological synchrony was also not found to interact with the pain difference to a statistically significant level ($B = 0.18$, $t(23) = 0.95$, $p = 0.353$, 95% CI[−0.22, 0.58]). This was not the case for the magnitude of G2 nocebo hyperalgesia on G3 nocebo hyperalgesia which reached statistical significance ($B = 0.47$, $t(25) = 2.80$, $p = 0.010$, 95% CI[0.12, 0.81]), and interacted with psychological synchrony ($B = 0.47$, $t(23) = 3.67$, $p = 0.001$, 95% CI[0.20, 0.73]). As presented in Fig. 4c, when the G2-Demonstrator expressed minimal nocebo hyperalgesia there was limited effect of psychological synchrony on the subsequent Demonstrator's (i.e., G3-Demonstrator's) nocebo effect. In this instance, there was presumably limited information regarding treatment-related pain to pass on, irrespective of how attentive the Observer was. Differences occurred at high levels of synchrony, where greater nocebo hyperalgesia at G2 generated greater nocebo hyperalgesia at G3. Full statistical information is presented in Supplementary Table 1. No statistically significant differences were found for indices of physiological synchrony (DET and LAM).

## Manipulation check 2: responders and non-responders

Full transmission of Nocebo Hyperalgesia was observed across the transmission chain, rather than being moderated through chain position to a statistically significant level (G2 > G3). Our original stats plan was to check for Responder-Status. However, given the strength of social transmission we present this information in Supplementary Note 3 and Supplementary Fig. 4. Of note, however, was that the proportion of Responders/Non-Responders did not differ at a statistically significant level across G2 and G3 groups ($\chi^2(1, N = 65) = 0.001$, $p = 0.975$, Cramer's $V = 0.004$) suggesting that there was a limited change in receptivity to the manipulation at later stages of the chain (all G1 participants were Responders). Further, Responders (collapsed across G1 and G2) were found to have significantly higher Expectancy ratings, FAU$_{heat}$, EDA$_{heat}$ and EDA$_{cue}$ scores to the Tx, relative to the NT cue (full statistics in Supplementary Note 3). This was not the case for Non-Responders, suggesting that responsiveness was not limited to self-report pain perception, but also manifested in physiological receptivity to the manipulation.

## Exploratory analysis

**Observer responses to demonstrators.** It is important to establish a complete understanding of the processes involved during the social transmission of pain, including from the perspective of both Demonstrators and Observers. Direct nociceptive pain experienced by the Demonstrators has been the focus of analysis regarding intrapersonal pain. However, little is known about the vicarious or 'empathetic pain' elicited in the Observers when watching the experience of pain[78]. To achieve this, an exploratory analysis was run concerning the Observer response in primary and secondary outcomes while watching the Demonstrator (see Fig. 5). In the case of anticipatory physiological outcomes, there was no statistically significant main effect of Cue for FAU$_{cue}$ ($F(1, 61) = 0.09$, $p = 0.761$, $\eta_p^2 < 0.01$, 95% CI[0.00, 0.07]) and EDA$_{cue}$ ($F(1, 62) = 3.53$, $p = 0.976$, $\eta_p^2 < 0.01$, 95% CI[0.00, 0.00]), nor were there statistically significant interactions between Cue and Generation (FAU$_{cue}$: $F(1, 61) = 0.61$, $p = 0.436$, $\eta_p^2 = 0.01$, 95% CI[0.00, 0.11] | EDA$_{cue}$: $F(1, 62) = 3.53$, $p = 0.065$, $\eta_p^2 = 0.05$, 95% CI[0.00, 0.19]). There was a statistically significant interaction between Cue and Generation on Expectancy ratings ($F(1, 58) = 5.98$, $p = 0.018$, $\eta_p^2 = 0.09$, 95% CI[0.00, 0.25]), however, the effect of Cue for both G2 ($F(1, 58) = 38.65$, $p < 0.001$, $\eta_p^2 = 0.04$, 95% CI[0.21, 0.55]) and G3 ($F(1, 58) = 5.83$, $p = 0.019$, $\eta_p^2 = 0.09$, 95% CI[0.00, 0.25]) reached statistical significance. With respect to physiological outcomes in response to watching the onset of the painful stimulus, there was a statistically significant interaction between Cue and Generation on FAU$_{heat}$ ($F(1, 62) = 4.45$, $p = 0.039$, $\eta_p^2 = 0.07$, 95% CI[0.00, 0.21]), where G2 significantly differed in their response to the Cue ($F(1, 62) = 26.37$, $p < 0.001$, $\eta_p^2 = 0.30$, 95% CI[0.12, 0.46]) but this did not reach a level of statistical significance for G3 ($F(1, 62) = 3.17$,

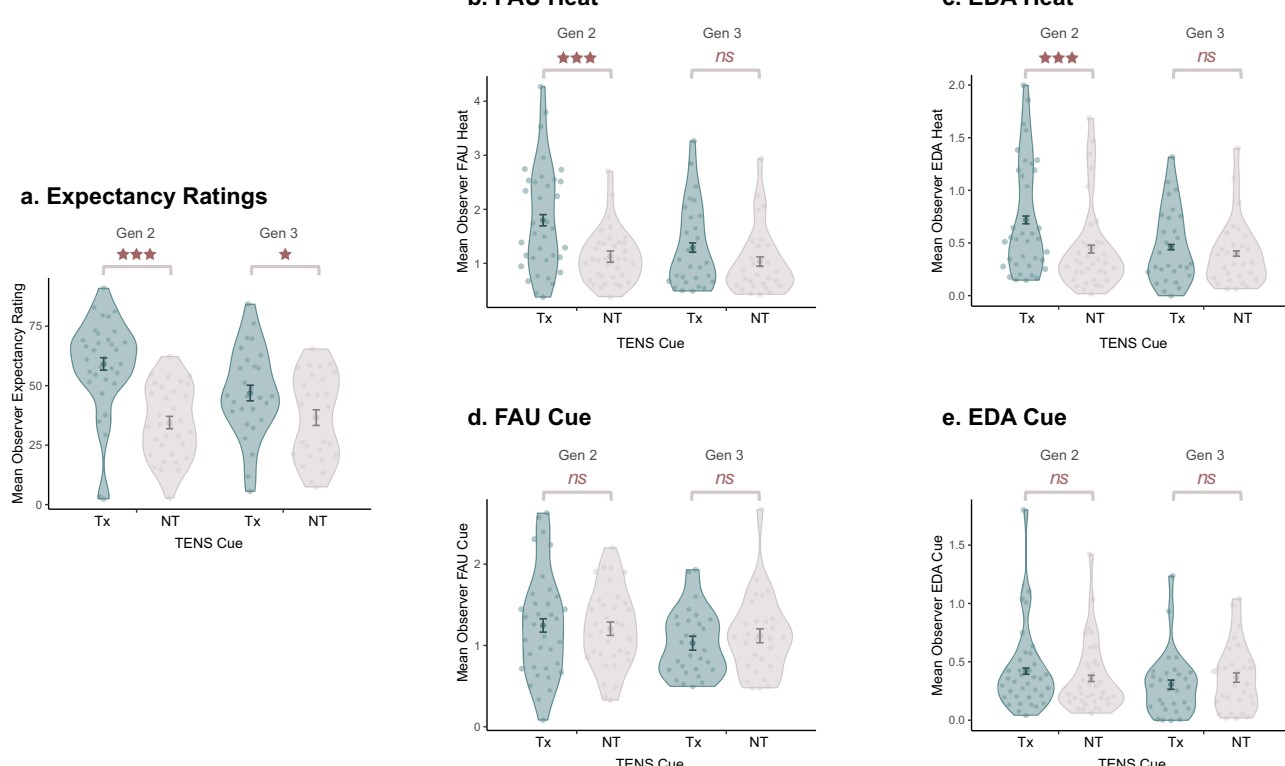

**Fig. 5 | Mean pain-related responses expressed by G2 and G3 observers when watching G1 and G2 demonstrators.** (**a**) Mean differences in Expectancy Ratings between TENS Treatment (Tx) and No Treatment (NT) trials for G2 ($N = 33$ biologically independent samples due to three missing values in Observer data) and G3 ($N = 28$ biologically independent samples due to one missing value in Observer data); (**b**) Mean differences in FAU Heat for G2 ($N = 35$ biologically independent samples) and G3 ($N = 29$ biologically independent samples); (**c**) Mean difference in EDA Heat for G2 ($N = 35$ biologically independent samples) and G3 ($N = 29$ biologically independent samples); (**d**) Mean difference in FAU Cue for G2 ($N = 36$ biologically independent samples) and G3 ($N = 28$ biologically independent samples due to one missing value in Observer data); (**e**) Mean difference in EDA Cue for G2 ($N = 35$ biologically independent samples) and G3 ($N = 29$ biologically independent samples). In all panels, for G2 Observers depicted, G1 Demonstrators experienced Tx trials paired with the High Temperature Destination (54 °C) and NT trials with the Low Temperature Destination (45 °C). For G3 Observers depicted, G2 Demonstrators experienced both Tx and NT trials paired with the Low Temperature Destination (45 °C). Figures are split by Generation. Error bars represent $+/- $ 1SED, adjusted for the within-subjects comparison (using the *afex* package in R[72]). Significance levels are depicted as (***$p < 0.001$); (**$p < 0.01$), and (*$p < 0.05$).

$p = 0.080$, $\eta_p^2 = 0.05$, 95% CI[0.00, 0.19]). A similar interaction applied to EDA$_{heat}$ ($F(1, 62) = 10.62$, $p = 0.002$, $\eta_p^2 = 0.15$, 95% CI[0.02, 0.31]) with a comparable pattern across Generation (G2: $F(1, 62) = 39.06$, $p < 0.001$, $\eta_p^2 = 0.39$, 95% CI[0.20, 0.54] | G3: $F(1, 62) = 1.53$, $p = 0.220$, $\eta_p^2 = 0.02$, 95% CI[0.00, 0.14]). However, the absence of a statistically significant effect of Cue at G3 may be explained by the presence of Non-Responders at G2 (see Supplementary Note 4).

**Observer responses on the subsequent expression of nocebo hyperalgesia.** Finally, we assessed how empathetic pain expressed by individual Observers predicted the magnitude of their own nocebo hyperalgesia when they subsequently became the Demonstrator. That is, how their response to one individual expressing pain modulated the way in which they transmitted pain-related information to the next person. To minimise the number of analyses, Expectancy Ratings, FAU$_{heat}$ and EDA$_{heat}$ were selected. This is because there was limited Observer response in FAU$_{cue}$ and EDA$_{cue}$ (see Fig. 5) and Observers did not provide Pain Ratings meaning that this variable could not be added to the model. Step 1 of the hierarchical regression analyses contained the Observer Response (difference score), Step 2 controlled for Generation (G2/G3), while Step 3 contained the Observer by Generation interaction. Full model statistics are presented in Supplementary Table 2.

The Observer's Expectancy Ratings interacted with Generation to predict the magnitude of subsequent nocebo hyperalgesia ($B = -0.15$,

$t(59) = 2.06$, $p = 0.044$, 95% CI[$-0.29$, $-0.01$]). Simple slopes revealed that the slope for G2-Observers did not reach statistical significance ($B = 0.08$ 95% CI[$-0.12$, 0.29]) while it did for G3 ($B = 0.37$ 95% CI[0.18, 0.57]). In summary, greater differences in expected pain when watching a Demonstrator receive treatment vs. no treatment predicted the magnitude of nocebo hyperalgesia among those watching a nocebo hyperalgesic response, but this association was reduced among those watching a pain response generated by differences in nociceptive intensity. Observer FAU$_{heat}$ (i.e., increased pain-related facial expressions pulled when watching the Demonstrator in pain) predicted subsequent nocebo hyperalgesia, independent of Generation ($B = 4.77$, $t(59) = 2.32$, $p = 0.024$, 95% CI[0.66, 8.88]). Observer EDA$_{heat}$ interacted with Generation ($B = -19.38$, $t(59) = 2.65$, $p = 0.010$, 95% CI[$-34.03$, $-4.73$]). The pattern of results was opposite to self-report Expectancy. While simple slopes did not reach a level of statistical significance for G3 ($B = 22.0$ 95% CI[$-3.68$, 47.69]), decreased autonomic responding to the Demonstrator increased the nocebo effect at G2 ($B = -16.7$ 95% CI[$-30.85$, $-2.65$]).

**Summary of results**

Figure 6 consolidates the evidence presented above by presenting a graphical summary of *inter*personal outcomes at the level of the dyad, and *intra*personal outcomes relating to the Observer's response to the Demonstrator, which went on to predict pain modulation to a statistically significant level in that same Observer when they subsequently Demonstrated. Differences in physiological response during Block 1 went on to predict the magnitude of

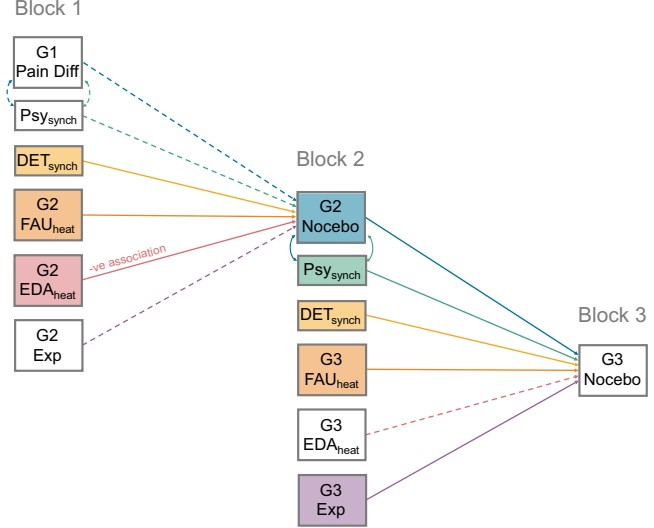

**Fig. 6 | Summary of results.** Summary of the *inter*personal outcomes between dyads, and *intra*personal outcomes at the level of the Observer, that went on to predict that magnitude of that Observer's nocebo effect in the subsequent block (i.e., when they Demonstrated). Please note that this Figure is for illustrative purposes with each regression path calculated separately (as presented in the analyses above) and not estimated as an overall path model. Filled boxes and arrows represent significant predictors of the subsequent Demonstrator's nocebo effect, while dashed arrows represent non-significant predictors. 'G2 Nocebo' ($N = 36$ biologically independent samples) and 'G3 Nocebo' ($N = 29$ biologically independent samples) boxes represent this nocebo effect (the difference between Tx ('Treatment') and NT ('No Treatment') Cues). This difference between Cues is denoted as a 'Pain Difference' at G1 ($N = 36$ biologically independent samples), due to the genuine difference in nociceptive stimuli. Recursive arrows between 'G2 Nocebo' and 'Psych$_{synch}$' boxes represent the significant interaction term between these factors when predicting the subsequent nocebo effect in the chain. Abbreviations are as follows: Psych$_{synch}$ = Psychological Synchrony; DET$_{synch}$ = Physiological Synchrony (Determinism CRQA metric); Exp = Expectancy Rating.

the Observer's nocebo effect during Block 2. However, During Block 2, the Observer's (G3) expectancy, as well as their ability to make predictions about the Demonstrators upcoming pain, in interaction with the magnitude of the G2 Demonstrator's nocebo effect, predicted nocebo hyperalgesia in the same Observer when they Demonstrated during Block 3. Results therefore highlight subtle differences in predictors of nocebo hyperalgesia elicited via watching a pain response generated by differences in nociceptive intensity vs. watching nocebo hyperalgesia.

## Discussion

A multi-generational social transmission chain was employed to explore whether witnessing treatment-related pain exacerbated subsequent pain in the observer, and whether this pain modulation could spread to an individual who never witnessed the original pain experience. Two primary results emerged. First, nocebo hyperalgesia passed between individuals. Second, this spread was exacerbated by interpersonal synchrony, demonstrating that synchronous interaction does not always result in positive outcomes[54,55]. This suggests that those closest to us may transmit their symptoms most readily[79], even in the case of the nocebo effect.

In relation to our pre-registered hypotheses, results demonstrated that genuine pain responses in G1-Demonstrators (supporting H[1]) caused modulation of subjective and physiological outcomes for otherwise undifferentiated thermal stimulation (i.e., nocebo hyperalgesia) in subsequent Generations (supporting H[2]). This modulation was stronger than expected as it propagated down the entire chain, with the magnitude of nocebo hyperalgesia in G2- and G3-Demonstrators not differing to a statistically significant level (contrary to H[3]). Physiological Synchrony (EDA) with the previous Demonstrator was found to predict nocebo hyperalgesia in both

Generations, however, Psychological Synchrony (Expectancy ratings) predicted this effect to a statistically significant level only in G3 (partially supporting H[4]). For G3, the effect of Psychological Synchrony on pain modulation was influenced by the magnitude of nocebo hyperalgesia expressed by G2, but there was no statistically significant evidence for this effect for Physiological Synchrony (partially supporting H[5] and indicating potential differences in the contribution of physiological and subjective self-report responses in social transmission).

The present study therefore extends the existing literature by investigating whether witnessing nocebo hyperalgesia can generate another nocebo hyperalgesic response in the Observer. Existing research has concerned discrete one-to-one social transmission that occurs on a limited one-off basis, with little indication of how the Observer's subsequent expression of pain might go on to influence others. This is with the exception of one study that investigated nausea, not pain[76]. As the expression of socially-modulated pain has been suggested to differ from that acquired via direct experience, both in terms of autonomic and brain activity[17,25], it was previously unclear whether phenomenological differences in social-induced nocebo hyperalgesia would limit its spread to others. Results presented here demonstrate that this is not the case, with nocebo hyperalgesia occurring in both G2 *and* G3. This indicates that transmission of negative expectancies need not require the observed treatment, such as a drug or intervention, to possess any active pain-inducing property. Socially-acquired pain modulation therefore has the potential to spread rapidly and perpetuate itself in real world settings, such as clinics or hospitals.

These results may be considered in light of the broader literature regarding social contagion[80]—the transfer of a behavioural or emotional response between conspecifics. Social contagion is generally considered to be adaptive, for example, aiding interpersonal connection and avoiding threats in the environment[7]. However, it can also lead to the development of maladaptive responses such as chronic stress[6]. Interpersonal synchrony may be one factor that facilitates this social transmission[8,81,82], but with respect to pain has only been demonstrated to confer empathetic and pro-social benefits in human participants[44–46]. For example, previous research has demonstrated that neural and physiological synchrony can attenuate the experience of pain[47,48,50–52]. However, our results demonstrate that physiological synchrony is not purely beneficial and can also play a maladaptive role in the subsequent experience of pain. Irrespective of Generation, those who demonstrated synchrony in their physiological responses went on to experience greater nocebo hyperalgesia themselves. Observer/Demonstrator synchrony in the ability to predict upcoming pain predicted nocebo hyperalgesia, but this only reached statistical significance among G3 participants. This effect was most pronounced when the previous Demonstrator displayed a larger nocebo hyperalgesic response. As outlined below, it is likely that the effect among G3 participants is a consequence of the Observer's predictions regarding the Demonstrator being more in line with their own experience when they underwent the thermal procedure. The implication being that, provided an observed painful experience is within a range applicable to one's own experience, the stronger its presentation, the larger the transmitted nocebo effect.

That similarity in experience may drive nocebo hyperalgesia is demonstrated by the prevalence of 'non-responders' across Generations. Non-responders are common in research concerning nocebo hyperalgesia and placebo analgesia, with some studies estimating as many as ~61–64% of participants in placebo analgesia and 44% of participants in nocebo hyperalgesia fail to exhibit a significant response[20,75]. In the current study, despite all witnessing a G1-Demonstrator experience 'treatment'-related pain, approximately half of G2 participants (56%) failed to show a statistically significant nocebo effect in their pain ratings. We could not conclude that this was a consequence of observing a G1-Demonstrator who expressed reduced treatment-related pain, as the G1 pain difference did not predict the strength of nocebo hyperalgesia in the subsequent participant to a statistically significant level. However, pain ratings were much higher among G1 participants than G2 (see Fig. 4a). Research has suggested that the

expression of the nocebo effect may be inhibited when the incoming pain signal is sufficiently discrepant from pre-existing expectations regarding that pain[83,84]. In our study, the nociceptive stimulus given to G2 may have been too low to induce significant nocebo hyperalgesia in certain participants who were expecting an extremely high level of pain given the responses they had observed in G1. In contrast, the proportion of non-responders stabilised between G2 and G3 (56% and 55%, respectively). This may be due to the greater similarity in the actual experience of the nociceptive stimulus between them. If so, this suggests that the strength of nocebo hyperalgesia socially transmitted from one individual to another may be relatively resistant to weakening once established. This would mean that such effects have the capability to propagate through many subsequent generations. However, the fact that this pattern of results was observed in pain ratings and psychological, but not physiological, synchrony is notable and suggests different pathways regarding self-report perceptions and autonomic responsivity, as has been suggested elsewhere[25,85].

An unusual pattern of results was found in which *increased* electrodermal activity while observing a dyad partner receive thermal stimulation was associated with a *lower* subsequent nocebo effect in G2, but this did not reach statistical significance in G3. There is some evidence suggesting that high levels of physiological arousal may impede social learning of nociceptive information[86]. However, post-hoc examination of the 'Reverse-Responders' data (Supplementary Note 3) provides an alternative explanation. On removing Reverse-Responders, EDA$_{heat}$ was no longer predictive of pain modulation in either generation to a level of statistical significance. In addition, Observer Expectancies were now significantly predictive of the subsequent nocebo effect across both generations, where previously this had been true for G3 only. This parallels our earlier work regarding the effect of direct conditioning on pain modulation, where expectancies formed when learning about a treatment predicted pain modulation at test[70]. The remaining pattern of results stayed the same, although becoming stronger in most instances, especially LAM (physiological synchrony) which reached statistical significance when Reverse-Responders were removed. It is unclear why some participants responded to the wrong treatment cue. One possibility is that social cues from the previous Demonstrator were more ambiguous, and therefore provided less information about which cue indicated the treatment. Further research is needed to understand the factors that cause a breakdown in the communication of pain-relevant information and to replicate the post-hoc results above. It is also unclear why EDA$_{heat}$ during observation did not cohere with FAU and physiological synchrony data in predicting the subsequent nocebo effect. Some research has demonstrated differences in facial expressions and EDA during observational and conditioned responding, suggesting that these metrics may be partially dissociable[45]. One explanation is that FAUs and physiological synchrony either form a direct visual display indicating to the Demonstrator that the Observer is mirroring them (i.e., FAUs) or can be triggered by these direct displays (e.g., changes in posture or breathing patterns over time; physiological synchrony). Future research should therefore determine whether outward displays of synchrony, which may serve to reinforce symbiotic responding, are more predictive of nocebo hyperalgesia. Finally, our results were concerned with the unidirectional propagation of pain down the social transmission chain (i.e., moving from G1 to finish G3). Given that synchrony is bi-directional in nature, it may be interesting for future research to investigate whether any information is transmitted from the observer to the demonstrator—potentially influencing future experiences of the demonstrator as well.

## Limitations

Some limitations in the present study should be noted. First, due to available resources, a confederate took the position of the final Observer ensuring that two people were always present in each block. This confederate was blind to the treatment cue. However, it possible that their presence may have influenced the social interaction in some subtle way. Future research should therefore replicate the present paradigm with a genuine participant in this position. In the present study, expectancy ratings during Observation concerned predictions about the Demonstrator's pain, but not necessarily whether the Observer would have expected the same amount of pain *themselves*. Conditioning paradigms have demonstrated that expectations regarding pain predict the magnitude of the nocebo effect[87]. However, this is often with respect to direct learning paradigms, where pain is delivered during acquisition meaning expectations are directly relevant to previous experience of pain. There are few social observation studies testing the effect of expectancy on pain. While all have enquired about the participant's own expected pain, this has occurred either retrospectively after experiencing pain[15], or in conjunction with a direct conditioning procedure[16,17]. Consequently, little is known about socially-acquired expectations prior to directly experiencing a painful stimulus. Given generational differences existed in expectancy metrics, future research should incorporate explicit measures regarding Observer's personal expectations for pain to better understand the conditions under which personal beliefs regarding upcoming pain may be violated.

## Conclusion

In summary, the present study demonstrates that nocebo hyperalgesia is insidious in that it can be propagated along a social chain and can be induced in contexts where the pain witnessed is driven by negative expectations only. This may underlie phenomena such as the spread of over-reporting vaccine-related side effects within a community, or other social contagion events[11,88]. Importantly, our results show that the degree to which nocebo hyperalgesia propagates is also affected by the degree of synchronicity between the people involved. These findings highlight the importance of considering interpersonal factors in any attempt to develop interventions to reduce socially-acquired nocebo hyperalgesia.

## Data availability

Individual data is available through the Open Science Framework (OSF) repository: https://osf.io/qzstu/?view_only=68a87230dcfe4245bb4dc5d0564cf0d9

## Code availability

R code necessary to replicate primary analyses is available through the Open Science Framework (OSF) repository: https://osf.io/qzstu/?view_only=68a87230dcfe4245bb4dc5d0564cf0d9

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

## Acknowledgements

This research was supported by Australian Research Council grants DE160100864 and DP180102061. The funding body had no involvement in study design, analysis, interpretation, writing, or the decision to submit the present article for publication.

## Author contributions

Author contributions are described using CRediT (Contributor Roles Taxonomy: https://credit.niso.org/). Conceptualization: K.B. and N.A.M.; Data curation: N.A.M.; Formal Analysis: K.B.; Funding acquisition and Resources: B.C.; Investigation: R.M., C.S., W.T.; Methodology: K.B., N.A.M., R.M., and B.C.; Software: K.B. and N.A.M.; Supervision: K.B.; Visualization: K.B.; Writing – original draft: R.M., K.B., and N.A.M.; Writing – review & editing: R.M., K.B., N.A.M., C.S., W.T., and B.C.

## Competing interests

The authors declare no competing interests.
