## [Peer Review File · Communications Psychology]

13th Sep 23

Dear Dr. Barnes,

Thank you for your patience during the peer-review process. Your manuscript titled "Your pain increases my pain: Exploring the effect of interpersonal synchrony on socially-acquired pain" has now been seen by 3 reviewers, and I include their comments at the end of this message. They find your work of interest, but raised some important points. We are interested in the possibility of publishing your study in Communications Psychology, but would like to consider your responses to these concerns and assess a revised manuscript before we make a final decision on publication.

We therefore invite you to revise and resubmit your manuscript, along with a point-by-point response to the reviewers. Please highlight all changes in the manuscript text file.

Editorially, we consider it important that you address the reviewers' request for additional clarifications regarding your methods and findings. Be sure to include a working link to your preregistration in the revised manuscript. Please do not remove any analyses that were preregistered.

We ask that you ensure your manuscript complies with our editorial policies. Please ensure that the following formatting requirements are met, and any checklist relevant to your research is completed and uploaded as a Related Manuscript file type with the revised article. In the Reporting Summary, you marked "doesn't apply" for reporting plot distributions, but it appears that your manuscript includes bar graphs. To ensure comprehensive reporting, please revisit this aspect and considering whether any further clarification or details about plot distributions can be included in your manuscript.

Editorial Policy: [Policy requirements](https://www.nature.com/documents/nr-editorial-policy-checklist.pdf) (Download the link to your computer as a PDF.) Please note that the manuscript does not currently contain an "ethics and inclusivity statement", which is not a problem, but the box should not be ticked.

Furthermore, please align your manuscript with our format requirements, which are summarized on the following checklist:

[Communications Psychology formatting checklist](https://www.nature.com/documents/commspsychol-style-formatting-checklist-article-rr.pdf)

and also in our style and formatting guide [Communications Psychology formatting guide](https://www.nature.com/documents/commspsychol-style-formatting-guide-accept.pdf) .

Please use the following link to submit your revised manuscript, point-by-point response to the referees' comments (which should be in a separate document to any cover letter) and the completed checklist:

[link redacted]

Please do not hesitate to contact me if you have any questions or would like to discuss these revisions further. We look forward to seeing the revised manuscript and thank you for the opportunity to review your work.

Best regards,

Yafeng Pan

Yafeng Pan, PhD
Editorial Board Member
Communications Psychology
orcid.org/0000-0002-5633-8313

EDITORIAL POLICIES AND FORMATTING

* **CODE AVAILABILITY:** All Communications Psychology manuscripts must include a section titled "Code Availability" at the end of the methods section. In the event of publication, we require that the custom analysis code supporting your conclusions is made available in a publicly accessible repository; at publication, we ask you to choose a repository that provides a DOI for the code; the link to the repository and the DOI will need to be included in the Code Availability statement. Publication as Supplementary Information will not suffice. We ask you to prepare code at this stage, to avoid delays later on in the process.

*** DATA AVAILABILITY:**

All Communications Psychology manuscripts must include a section titled "Data Availability" at the end of the Methods section or main text (if no Methods). More information on this policy, is available at <http://www.nature.com/authors/policies/data/data-availability-statements-data-citations.pdf>.

At a minimum the Data availability statement must explain how the data can be obtained and whether there are any restrictions on data sharing. Communications Psychology strongly endorses open sharing of data. If you do make your data openly available, please include in the statement:

We recommend submitting the data to discipline-specific, community-recognized repositories, where possible and a list of recommended repositories is provided at <http://www.nature.com/sdata/policies/repositories>.

If a community resource is unavailable, data can be submitted to generalist repositories such as [figshare](https://figshare.com/) or [Dryad Digital Repository](http://datadryad.org/). Please provide a unique identifier for the data (for example a DOI or a permanent URL) in the data availability statement, if possible. If the repository does not provide identifiers, we encourage authors to supply the search terms that will return the data. For data that have been obtained from publicly available sources, please provide a URL and the specific data product name in the data availability statement. Data with a DOI should be further cited in the methods reference section.

REVIEWERS' EXPERTISE:

Reviewer #1 pain/empathy for pain
Reviewer #2 synchrony/social cognition
Reviewer #3 social learning/synchrony

REVIEWERS' COMMENTS:

Reviewer #1 (Remarks to the Author):

I am an expert on pain and empathy for pain, and I work on placebo and nocebo effects, but I have no experience with FAC and EDA data or synchrony analysis.

Summary

In general, the main question of the present manuscript was to investigate how pain perception and experiences with certain treatments (thus expectations and experiences) transmit through generations of participants (G1-G4). The authors created a sham placebo treatment in G1 (high pain), which was then observed and demonstrated in the next generation without the treatment (low pain). The authors report that placebo effects were transferred from G1 to the other generations and the degree of synchrony in skin conductance and pain prediction predicted the degree of this transfer.

Strengths

The research question is fascinating and relevant, I was really excited to read this paper and it did not disappoint, this paper was a pleasure to read! The manuscript is well structured and easy to follow. The authors provide a strong rationale for their work and clearly derived hypotheses. I highly appreciate that the authors employed a range of methods to tackle their research question, and a range of statistical techniques and analyses to gather multiple parts of evidence. I really like the fact that the authors nearly only used real participants and minimal confederates, thereby increasing the amount and usefulness of the collected data. Methods are very detailed and allow for replication. Results are well described, follow a logical story, and relevant statistics are reported. I appreciate the clear figures and especially Figure 5 that summarizes all findings. I also appreciate the transfer of the authors findings to real world examples in the discussion. The authors answered nearly every open question I had when reading the paper.

Nevertheless, I have some reservations, as outlined below, which need to be addressed, before I can recommend the article to be of sufficient quality to be published. My comments are structured as they appear in the text. I hope the authors find my suggestions helpful to improve the manuscript.

Comments

Line 4 speaks of G1-G3, but later in the abstract there is also G4, maybe clarify to avoid confusion? In line 73 onwards, I would also mention G4 then? I understood only in line 88 that G4 is a confederate, so this needs to be made more explicit. Maybe also in the subtitle of Figure 1?

The authors investigated the effect of a negative treatment, which leads to placebo hyperalgesia, yet the title and abstract never once mention the word placebo. I think it would be beneficial to frame the work in light of negative expectations and placebo effects for readers already early in the abstract to avoid misunderstanding.

A recent meta-analysis on social observational learning came out that the authors might want to incorporate (just a suggestion, only if it fits)

https://journals.lww.com/pain/Fulltext/9900/Learning_pain_from_others__a_systematic_review_and.328.aspx

Along the same lines, the key Benedetti paper might also be interesting, as it also investigated transmission of negative information and symptoms among different people? (again, just a suggestion) <https://www.sciencedirect.com/science/article/abs/pii/S0304395914000244>

Were the heat stimuli individually calibrated, as line 77 indicates a fixed temperature for high and low stimuli? 45 degrees Celsius is not a low intensity stimulus, it can actually be very painful for some people. Why did the authors choose to not use individuals pain levels and could this have influenced their results?

Please explain already at the start of the results, why “trial” was additionally included as a factor.

Line 95 psychological synchrony should be lower-case letters?

Out of curiosity and as a second manipulation check, I would also be interested whether the authors ran comparisons of their outcome measures with G1? I see comparisons between G2 and G3, but it would also be interesting how the nocebo effect differs from the experienced treatment effect in G1?

Figure 3c: Is there a typo? In the title it says Nocebo (G2) but the Figure shows Nocebo (G3) and also lines for Nocebo (G2).

Did the authors check whether there was a difference between generations in the matching of Responder and Nonresponder dyads?

Is this high rate of > 50% non-responders common regarding previous own and others' work? It might be helpful to put these numbers into perspective.

Table 1: Please explain the abbreviations DET and LAM in the Table itself.

The fact that the authors preregistered the study only becomes apparent at the end, in the methods, I think this can already be highlighted much sooner Please also include the link to the preregistration directly in the paper.

Could the confusion of the cues have been due to color-blindness (Tritanomaly or Tritanopia)? Did the authors check for these issues/were the cues clearly separable?

What role do the authors think that gender (and matching of gender between observer and demonstrator) played in their study and findings?

In the discussion, it might potentially be interesting to talk about bidirectional influences, so whether this chain can also reverse and influence future responses of the earlier chain members (considering the fact that synchrony is not unidirectional)? Just an idea :)

The authors also measured confidence in regard to the cues. Did I miss it are the results of these ratings reported somewhere?

References

Benedetti, F., Durando, J., & Vighetti, S. (2014). Nocebo and placebo modulation of hypobaric hypoxia headache involves the cyclooxygenase-prostaglandins pathway. *PAIN*[®], 155(5), 921-928.
Meeuwis, Stefanie H.; Wasylewski, Mateusz T.; Bajcar, Elżbieta A. Bieniek, Helena; Adamczyk, Wacław M.; Honcharova, Sofiiia; Di Nardo, Mariannac; Mazzoni, Giulianac; Bąbel, Przemysława. Learning pain from others: a systematic review and meta-analysis of studies on placebo hypoalgesia and nocebo hyperalgesia induced by observational learning.
PAIN:10.1097/j.pain.0000000000002943

Reviewer #2 (Remarks to the Author):

This paper examined the effects of perceived pain on one's own pain experience, how this is passed down through generations, and modulated by synchrony (both physiological and behavioral) between the demonstrator and observer. I found the paper to be well-written, intriguing, and innovative.

One of the most interesting findings was the evident transmission of nocebo pain up to the third generation. Below are several points that I believe could enhance this manuscript and connect it to more extensive work beyond the realm of nocebo effects.

Abstract – Upon my first reading, I must admit that I found the abstract rather puzzling, understanding it more clearly only after reading the entire paper. It begins with, “Witnessing another's pain can intensify pain in the observer. However, current research has concentrated on the observer's intrapersonal experience, overlooking social reciprocity.” In discussing reciprocity, I expected the authors to examine the impact of the observer on the demonstrator (effects of G2 on G1) rather than how the effect disseminates to other members. The term “generations” in the abstract was also confusing, as I initially believed it referred to family generations or age variations.

Introduction: The section on nocebo is clear and informative. Nonetheless, presenting this solely in the context of nocebo effects neglects a vast realm of research on the evolutionary facets of social contagion and learning, including those specific to pain, stress, or fear, which could be responsible for these “side-effects”. For instance, observing another's pain and subsequently experiencing or simulating it is considered fundamental for empathy and for learning to avoid potentially painful or stressful situations. In extreme scenarios (e.g., chronic stress or nocebo effects), this might have adverse impacts on the learner, yet it is generally an adaptive (and widely studied) mechanism.

Here's a recent paper on rodents that also reviews other relevant studies (including human ones): Keysers, C., & Gazzola, V. (2023). Vicarious emotions of fear and pain in rodents. *Affective Science*, 1-10.

And some additional relevant papers:

Brandl, H. B., Pruessner, J. C., & Farine, D. R. (2022). The social transmission of stress in animal collectives. *Proceedings of the Royal Society B*, 289(1974), 20212158.

Ashby, D., & McGirr, A. (2020). Direct Social Transmission of Chronic Stress. *Biological Psychiatry*, 87(9), S115.

Discussion: Similar to the Introduction, while the paper predominantly focuses on nocebo effects, the impacts should be considered in the broader context of emotion contagion, especially concerning pain.

Methods:

Preregistration: I was unable to locate the preregistration on the OSF, making it impossible to compare the analyses.

How was the participant number determined?

The term Psychological Synchrony might be somewhat confusing, as synchrony often suggests continuous temporal dynamics (as the authors themselves concede). A more accurate term would be more fitting. Perhaps terms like "Expectation of Pain Accuracy" or "Prediction of Pain" would be more appropriate.

Reviewer #3 (Remarks to the Author):

This paper investigates multi-generational social transmission of nocebo hyperalgesia. In particular it investigates if nocebo hyperalgesia can be transmitted across generational chains of participants in the presence of sham treatments and mechanistically if psychological or physiological synchrony between demonstrator and observer contributes to such transmission. The results support both these effects. The study is well-designed and analyses are appropriate. Interpretations of the results are in line with the data presented. I also appreciated the strong open science practices followed. In sum, this is a very good paper and I have only some minor comments, mostly suggestions for clarification. Thanks to the author for a fascinating read!

The methods were very clear both with regard to procedure and hypotheses. Unfortunately, I didn't find the description in section 2.1 as clear. I think information about what dyad pairs are the target of analysis could come earlier, together with hypothesis. I also first found (both here and in abstract) the information that there are 3 generations but a G4 participant somewhat confusing. Consider revisiting this section.

Please clarify in Figure that G4 is confederate.

In Figure 3, please add "psychological" to synchrony label on the x-axis

Maybe I missed it, but for psychological synchrony (correlation between expectancy ratings) what were the actual values here? How good were observers at predicting how much pain demonstrators thought they would be in?

For psychological synchrony, why is the demonstrator's pain expectancy rating used and not their actual pain? Unless I missed it the argument isn't spelled out in the paper and it would be useful to have it there.

I don't think it is necessary to present the step-wise regression, and variable selection by step-wise methods are generally not recommended. Instead, just present the full model (synchrony, generation, sync X generation), for each of the three synchrony measures.

Lines 248-249 - is it possible to explain in greater detail why the analyses concern the expression of empathic pain? Additionally why this is significant to explore.

For Figure 5, please add a figure legend for the meaning of the filled and dashed lines (non-significant and significant?).

19th October 2023

Communications Psychology

Re: Revision to the manuscript “Your pain increases my pain: Exploring the effect of interpersonal synchrony on the transmission of nocebo hyperalgesia”

Note for all Reviewers:

Please note that we have re-formatted the manuscript to reflect the formatting of Communications Psychology. The primary change being a switch to Methods-Results-Discussion formatting, rather than Results-Discussion-Methods. We have attempted to maintain consistency between the two versions in terms of overall content. We would like to thank all reviewers for taking the time to leave comments and suggestions that have improved the manuscript.

REVIEWERS' COMMENTS:

Reviewer #1 (Remarks to the Author):

I am an expert on pain and empathy for pain, and I work on placebo and nocebo effects, but I have no experience with FAC and EDA data or synchrony analysis.

Summary

In general, the main question of the present manuscript was to investigate how pain perception and experiences with certain treatments (thus expectations and experiences) transmit through generations of participants (G1-G4). The authors created a sham nocebo treatment in G1 (high pain), which was then observed and demonstrated in the next generation without the treatment (low pain). The authors report that nocebo effects were transferred from G1 to the other generations and the degree of synchrony in skin conductance and pain prediction predicted the degree of this transfer.

Strengths

The research question is fascinating and relevant, I was really excited to read this paper and it did not disappoint, this paper was a pleasure to read! The manuscript is well structured and easy to follow. The authors provide a strong rationale for their work and clearly derived hypotheses. I highly appreciate that the authors employed a range of methods to tackle their research question, and a range of statistical techniques and analyses to gather multiple parts of evidence. I really like the fact that the authors nearly only used real participants and minimal confederates, thereby increasing the amount and usefulness of the collected data. Methods are very detailed and allow for replication. Results are well described, follow a logical story, and relevant statistics are reported. I appreciate the clear figures and especially Figure 5 that summarizes all findings. I also appreciate the transfer of the authors findings to real world examples in the discussion. The authors answered nearly every open question I had when reading the paper. Nevertheless, I have some reservations, as outlined below, which need to be addressed, before I can recommend the article to be of sufficient quality to be published. My comments are structured as they appear in the text. I hope the authors find my suggestions helpful to improve the manuscript.

Comments

1. *Line 4 speaks of G1-G3, but later in the abstract there is also G4, maybe clarify to avoid confusion? In line 73 onwards, I would also mention G4 then? I understood only in line 88 that G4 is a confederate, so this needs to be made more explicit. Maybe also in the subtitle of Figure 1?*

We have reordered the abstract and have made clear that G4 is a confederate. We have also updated the caption of Figure 1 to reflect this. The description of the procedure in the introduction (Lines 68-93) has been updated to clarify the roles of each generation of participants and the purpose of the confederate.

2. *The authors investigated the effect of a negative treatment, which leads to nocebo hyperalgesia, yet the title and abstract never once mention the word nocebo. I think it would be beneficial to frame the work in light of negative expectations and nocebo effects for readers already early in the abstract to void misunderstanding.*

Thanks for this comment. We have updated the title to “You pain increases my pain: Exploring the effect of interpersonal synchrony **on the transmission of nocebo hyperalgesia**” to make this point clearer and now mention the nocebo effect in the abstract.

3. *A recent meta-analysis on social observational learning came out that the authors might want to incorporate (just a suggestion, only if it fits)*
https://journals.lww.com/pain/Fulltext/9900/Learning_pain_from_others__a_systematic_review_and.328.aspx

Along the same lines, the key Benedetti paper might also be interesting, as it also investigated transmission of negative information and symptoms among different people? (again, just a suggestion)

<https://www.sciencedirect.com/science/article/abs/pii/S0304395914000244>

Many thanks for these recommendations. We read the meta-analysis with interest and have included it in the first paragraph of the introduction (please see lines 34-35). We are aware of the Benedetti paper but decided not to include it due to limited word count. Our reasoning was that the study does not focus on experimentally induced pain per se, which was the focus of the literature review, but also involves transmission from one individual to many rather than along chains of individuals (social transmission, as studied here).

4. *Were the heat stimuli individually calibrated, as line 77 indicates a fixed temperature for high and low stimuli? 45 degrees Celsius is not a low intensity stimulus, it can actually be very painful for some people. Why did the authors choose to not use individuals pain levels and could this have influenced their results?*

We based the design of this study on a previously published paradigm (Barnes et al., 2021) that we are currently employing in multiple other studies. We have found that direct conditioning and social observation with the two temperatures selected produces a robust nocebo effect during test (i.e., at the lower temperature) in self-report ratings, but importantly, in facial action units and electrodermal activity, which the present study relies on. As such, we reasoned that the paradigm would sufficiently generate pain-related information at G1 to pass down the social chain. In studies where we have used calibration, we have found modulation of the autonomic response to be variable even during conditioning, which we assume is due to rapid nociceptor fatigue associated with painful stimuli, sometimes even at the upper temperatures used here (e.g., Greffrath et al., 2007). We therefore suspect that calibrating participants may weaken our effect. However, our focus is on relative within-subject changes between Tx and NT stimuli and we found no evidence to suggest that the 45°C temperature was experienced as particularly painful by participants. For ethical reasons, we employ a stringent thermal familiarisation procedure where all participants sample the full range of temperatures (up to 54°C) to take part in the study. All participants tolerated the stimuli well, even the upper temperatures, and none withdrew from the study.

5. *Please explain already at the start of the results, why “trial” was additionally included as a factor.*

We have now restructured the manuscript so that the methods section appears before the results. We have added a sentence to the primary analysis section of the methods to clarify why trial was included in the model (see lines 393-397):

Trial number was included in the analysis to assess extinction of the nocebo effect.

6. *Line 95 psychological synchrony should be lower-case letters?*

Due to reformatting, this text explaining the design of the study has been replaced with the Materials section.

7. *Out of curiosity and as a second manipulation check, I would also be interested whether the authors ran comparisons of their outcome measures with G1? I see comparisons between G2 and G3, but it would also be interesting how the nocebo effect differs from the experienced treatment effect in G1?*

Apologies if we have misinterpreted this question, but we did not measure the nocebo effect in G1 Demonstrators. The reason being that employing a test phase for these participants would have broken the flow of the transmission chain. Instead, G1 participants were comparable to social models in previous studies, but without an awareness of the manipulation. However, it would be interesting to determine whether the magnitude of the conditioned effect among G1 differs from that of social modelling. Unfortunately, this is beyond the scope of the present study. Our alternative interpretation of this question is whether the magnitude of the pain response at G1 predicted the nocebo effect at G2 which it did not (see 3.4.4. Interaction between Interpersonal Synchrony and the Pain Response expressed by the previous Demonstrator). We also depict the pain difference between Tx and NT stimuli in Figure 4a, which demonstrates that this difference was understandably much larger among G1 participants compared to G2 and G3.

8. *Figure 3c: Is there a typo? In the title it says Nocebo (G2) but the Figure shows Nocebo (G3) and also lines for Nocebo (G2).*

It is not a typo, Figure 3c (now 4c) depicts the interaction between the Nocebo effect of G2 and psychological synchrony on the nocebo effect of G3. We have amended the Figure caption to try and make this clearer.

Figure 4: 4a. represents the difference in pain ratings between Tx and NT trials for all Generations (error bars are +/- 1SD, not SEM), please note that while the y-axis is labelled 'nocebo hyperalgesia', G1 represents a pain response to direct nociception; 4b. depicts the Generation by Psychological Synchrony interaction on nocebo hyperalgesia; 4c. depicts the interaction between G2 nocebo hyperalgesia (represented as slopes at +/-1SD and the mean) and Psychological Synchrony (G2/G3; x-axis) on G3 nocebo hyperalgesia (y-axis).

9. *Did the authors check whether there was a difference between generations in the matching of Responder and Nonresponder dyads?*

All participants who were G1 Demonstrators were Responders. We checked whether the frequency of Responders and Non-Responders changed across Generations 2 and 3 (see Lines 567-570):

Of note, however, was that the proportion of Responders/Non-Responders did not differ significantly across G2 and G3 groups ($\chi^2(1, N=65)=0.001, p=.975$, Cramer's $V=.004$) suggesting that there was limited change in receptivity to the manipulation at later stages of the chain (all G1 participants were Responders).

We did not explore the data beyond this. The primary reason being the large number of pre-registered analyses in the manuscript and the fact that transmission of nocebo hyperalgesia was found independent of Responder status at G2 and G3. A secondary reason is that our permutation analysis is not set up to specifically answer this question as it does not account for the trajectory of responses over time. For example, Non-Responders may have rapidly extinguished, but generated enough social information at the beginning of the block to elicit a response in the next participant. We think this would be a great question for future research but would likely require a different design.

10. *Is this high rate of > 50% non-responders common regarding previous own and others' work? It might be helpful to put these numbers into perspective.*

We are not aware of many studies that report response rates. However, previous research has suggested that around 36-39% of participants respond to placebo manipulations, while around 56% respond to nocebo manipulations. As such, the numbers in the present study appear comparable. We have added text to the discussion to highlight this point (please see lines 722-725):

That similarity in experience may drive nocebo hyperalgesia is demonstrated by the prevalence of ‘non-responders’ across Generations. Non-responders are common in research concerning nocebo hyperalgesia and placebo analgesia, with some studies estimating as many as ~61 – 64% of participants in placebo analgesia and 44% of participants in nocebo hyperalgesia failing to exhibit a significant response^{20,71}. In the current study, approximately half of G2 participants (56%) failed to show a nocebo effect in their pain ratings, despite all witnessing a G1-Demonstrator experience ‘treatment’-related pain.

11. *Table 1: Please explain the abbreviations DET and LAM in the Table itself.*

The table has been updated to say “**Determinism (DET – recurrence in the EDA response)**” and “**Laminarity (LAM – stability in the EDA response)**”

12. *The fact that the authors preregistered the study only becomes apparent at the end, in the methods, I think this can already be highlighted much sooner Please also include the link to the preregistration directly in the paper.*

We have now edited the manuscript so that the methods section appears before the results, meaning this information no longer appears at the end. We have added a link to the pre-registration and have also uploaded a pdf version to OSF.

13. *Could the confusion of the cues have been due to color-blindness (Tritanomaly or Tritanopia)? Did the authors check for these issues/were the cues clearly separable?*

Participants were screened to ensure that they had normal or corrected-to-normal vision (we realise that this information was missing from the original version of the manuscript and have now included it – lines 107-108). Some estimates regarding the prevalence rates of tritanopia are ~1/14,000 and ~1/1,000,000 for tritanomaly, meaning that the chances of one of our subjects being affected is low. As such, we don’t believe that colour-blindness was a systematic confound in the present study. Robust conditioning effects have been demonstrated in previous studies using identical cues (e.g., Barnes et al., 2021), with clear differences in participant expectancy ratings between conditions, suggesting that the cues can be clearly distinguished by participants.

14. *What role do the authors think that gender (and matching of gender between observer and demonstrator) played in their study and findings?*

We pre-registered gender-match as a covariate based on previous research that suggests a potential role of participant gender in social modelling research. We did not observe any systematic effect of this covariate over the primary analyses and believe that other factors, such as interpersonal connection as studied here, may be stronger predictors. Other than ‘gender-match’, we did not look for differences in self-report gender across our analyses and are cautious of doing so. There are many different gender identities which can fluctuate over time. There are also multiple other features at the level of an individual that could generate stronger or weaker nocebo effects beyond gender which are difficult to control for in this type of research.

15. *In the discussion, it might potentially be interesting to talk about bidirectional influences, so whether this chain can also reverse and influence future responses of the earlier chain members (considering the fact that synchrony is not unidirectional)? Just an idea :)*

This is an interesting idea, which we have added to the Discussion section (see Lines 756-761):

Finally, our results were concerned with the unidirectional propagation of pain down the social transmission chain (i.e., moving from G1 to finish G3). Given that synchrony is bi-directional in nature, it may be interesting for future research to investigate whether any information is transmitted from the observer to the demonstrator—potentially influencing future experiences of the demonstrator as well.

16. The authors also measured confidence in regard to the cues. Did I miss it are the results of these ratings reported somewhere?

No, you did not miss this information. There was such a large amount of data that we felt this variable added little to the overall conclusions, given that there was evidence of transmission even with the non-responders in the sample. We originally included the confidence measure as a manipulation check in the eventuality that evidence for the transmission of the nocebo effect was not found. In case the data is of interest to readers, we have now included it in the Supplementary Materials where we talk about differences between G2/G3 Responders and Non-Responders in more detail.

There was an effect of Responder on TENS Confidence ($F(1, 62)=4.09, p=.022, \eta_p^2=.12$). Responders had higher confidence ($M=5.13, SE=0.31, 95\%CI [4.53, 5.75]$) than Non-Responders ($M=3.89, SE=0.31, 95\%CI [3.27, 4.52]$) when identifying the TENS Cue ($p=.016$). This result seems intuitive as Non-Responders reported experiencing a limited difference in pain between conditions. Reverse-Responders ($M=4.63, SE=0.58, 95\%CI [3.46, 5.79]$) did not differ from Responders or Non-Responders (both $ps>.50$).

References

- Benedetti, F., Durando, J., & Vighetti, S. (2014). Nocebo and placebo modulation of hypobaric hypoxia headache involves the cyclooxygenase-prostaglandins pathway. *PAIN*, 155(5), 921-928.
- Meeuwis, Stefanie H.; Wasylewski, Mateusz T.; Bajcar, Elżbieta A. Bieniek, Helena; Adamczyk, Wacław M.; Honcharova, Sofiiia; Di Nardo, Mariannac; Mazzoni, Giulianac; Bąbel, Przemysława. Learning pain from others: a systematic review and meta-analysis of studies on placebo hypoalgesia and nocebo hyperalgesia induced by observational learning. *PAIN*:10.1097/j.pain.0000000000002943

Reviewer #2 (Remarks to the Author):

This paper examined the effects of perceived pain on one's own pain experience, how this is passed down through generations, and modulated by synchrony (both physiological and behavioral) between the demonstrator and observer. I found the paper to be well-written, intriguing, and innovative.

One of the most interesting findings was the evident transmission of nocebo pain up to the third generation. Below are several points that I believe could enhance this manuscript and connect it to more extensive work beyond the realm of nocebo effects.

Abstract – Upon my first reading, I must admit that I found the abstract rather puzzling, understanding it more clearly only after reading the entire paper. It begins with, “Witnessing another's pain can intensify pain in the observer. However, current research has concentrated on the observer's intrapersonal experience, overlooking social reciprocity.” In discussing reciprocity, I expected the authors to examine the impact of the observer on the demonstrator (effects of G2 on G1) rather than how the effect disseminates to other members. The term “generations” in the abstract was also confusing, as I initially believed it referred to family generations or age variations.

Explaining the complexity of the design was difficult with such a limited wordcount. We have reordered the abstract and removed reference to social reciprocity, which we agree may be confusing in this instance. In reordering, we have also made clear that participants were randomised to ‘Generation’ to minimise confusion with respect to generations of family members. Based on the suggestion of R3, we have additionally highlighted that the G4-Observer was a confederate.

Introduction: The section on nocebo is clear and informative. Nonetheless, presenting this solely in the context of nocebo effects neglects a vast realm of research on the evolutionary facets of social contagion and learning, including those specific to pain, stress, or fear, which could be responsible for these “side-effects”. For instance, observing another's pain and subsequently experiencing or simulating it is considered fundamental for empathy and for learning to avoid potentially painful or stressful situations. In extreme scenarios (e.g., chronic stress or nocebo effects), this might have adverse impacts on the learner, yet it is generally an adaptive (and widely studied) mechanism.

Here's a recent paper on rodents that also reviews other relevant studies (including human ones):

Keyesers, C., & Gazzola, V. (2023). Vicarious emotions of fear and pain in rodents. *Affective Science*, 1-10.

And some additional relevant papers:

Brandl, H. B., Pruessner, J. C., & Farine, D. R. (2022). The social transmission of stress in animal collectives. *Proceedings of the Royal Society B*, 289(1974), 20212158.

Ashby, D., & McGirr, A. (2020). Direct Social Transmission of Chronic Stress. *Biological Psychiatry*, 87(9), S115.

Discussion: Similar to the Introduction, while the paper predominantly focuses on nocebo effects, the impacts should be considered in the broader context of emotion contagion, especially concerning pain.

Many thanks for this suggestion and the recommended papers. The links between social contagion and the nocebo effect as studied in the current manuscript are clear. We have updated the introduction to include reference to this research (see Lines 24-36):

Pain is a complex and near universal experience that involves psychological, cognitive, and social components¹⁻³. Even though we often learn about pain through others, the influence of social factors remains understudied^{4,5}. **Understanding the impact that social interaction has on pain is important. Research concerning both human and non-human animals has demonstrated that exposure to stress in others, including those experiencing pain, plays an important role when appraising and navigating the environment^{6,7}. This type of social exposure allows us to understand and empathise with the experiences of others⁸ but can also signal the presence of noxious stimuli or other environmental risk factors⁹. However, while often adaptive, our propensity to learn from others can have unwanted consequences. The nocebo effect, a pervasive problem where negative expectancies amplify symptoms such as pain (termed nocebo hyperalgesia), concerns one such instance. While most research has focused on key mechanisms of nocebo hyperalgesia that are non-social, growing evidence indicates that social learning plays a significant role in generating maladaptive pain experiences^{10,11}.**

We now also introduce social contagion when appraising our results concerning interpersonal synchrony in the Discussion (see Lines 694-701):

These results may be considered in light of the broader literature regarding social contagion⁷⁸, the transfer of a behavioural or emotional response between conspecifics. Social contagion is generally considered to be adaptive, for example aiding interpersonal connection and avoiding threats in the environment⁷. However, it can also lead to the development of maladaptive responses such as chronic stress⁶. Interpersonal synchrony may be one factor that facilitates this social transmission^{8,79,80}, but with respect to pain has only been demonstrated to confer empathetic and pro-social benefits in human participants⁴⁴⁻⁴⁶. For example, previous research has demonstrated that neural and physiological synchrony can attenuate the experience of pain^{47,48,50-52}. However, our results are the first to demonstrate that physiological synchrony is not purely beneficial and can play a maladaptive role in the subsequent experience of pain.

Given limitations on space, or preference is to focus on nocebo hyperalgesia for the rest of this section, given that this is what we set out specifically to test.

Methods:

Preregistration: I was unable to locate the preregistration on the OSF, making it impossible to compare the analyses.

Apologies, we thought that the pre-registration was uploaded. We have now included a link to the pre-registration in the paper and have also uploaded a pdf version to OSF.

How was the participant number determined?

Given the novelty of the procedure, we had no existing data regarding the expected effect size for the nocebo effect among G3 participants. However, we did have pilot data from a social modelling study in EEG using the same temperature stimuli. This provided an estimate for the effect at G2. As we thought that the nocebo effect

may reduce across generations, we were conservative and rounded then doubled the estimated sample size. Our total sample size is outlined in our pre-registration:

We found a large social-modelling effect size in a previous study (#52363): Cohen's $d_z = 1.67$ (Cohen's $f = 0.835$) which would result in a total sample size of 16 in each generation for the interaction term ($\alpha=0.5$; $\text{power}=90\%$). As we expect a reduction in the size of the socially modelled nocebo effect by position in the chain, we rounded and doubled this estimate to 30 participants in each condition. As some participants may withdraw from the testing session, we will test until there are 30 participants at G3.

The term Psychological Synchrony might be somewhat confusing, as synchrony often suggests continuous temporal dynamics (as the authors themselves concede). A more accurate term would be more fitting. Perhaps terms like "Expectation of Pain Accuracy" or "Prediction of Pain" would be more appropriate.

We agree with this point regarding the temporal characteristics of the EDA and expectancy outcomes (with expectancy ratings only being continuous at the level of the trial). We employed the term 'synchrony' across the EDA and expectancy measures to broadly denote that we were taking about intercorrelations at the level of the dyad for both outcomes. We really like 'pain prediction' as alternative terminology. However, reflecting on this change to the manuscript in conjunction with our pre-registration, we realised that readers may erroneously believe that we made no specific hypotheses about observer/demonstrator expectancy ratings because we use 'synchrony' to refer to both outcomes (EDA and expectancy) throughout. For consistency between documents, our preference would be to retain the term 'psychological synchrony'. However, we have added the following text to make clear that we are not talking about continuous temporal dynamics (see lines 343-349):

We note that this measure of 'Psychological Synchrony' differs from physiological measures of synchrony as the expectancy metric was only continuous at the level of the trial. Instead, in the present study, we use the terms physiological and psychological synchrony to broadly refer to intercorrelations between participants at the level of the individual dyads. As timeseries data at the trial level was not available (i.e., there was only one expectancy measure per trial), correlation coefficients were calculated for each dyad to represent the extent of synchronous responding.

We also make this clear again on lines 535-539:

One difference between physiological and psychological measures of synchrony (as presented above) is that the physiological metrics are calculated across the entire time course of the experimental block and are therefore not tied in a phasic manner to the presentation of the thermal stimuli, whereas the Expectancy ratings are specific to the upcoming stimulus on each trial.

Reviewer #3 (Remarks to the Author):

This paper investigates multi-generational social transmission of nocebo hyperalgesia. In particular it investigates if nocebo hyperalgesia can be transmitted across generational chains of participants in the presence of sham treatments and mechanistically if psychological or physiological synchrony between demonstrator and observer contributes to such transmission. The results support both these effects. The study is well-designed and analyses are appropriate. Interpretations of the results are in line with the data presented. I also appreciated the strong open science practices followed. In sum, this is a very good paper and I have only some minor comments, mostly suggestions for clarification. Thanks to the author for a fascinating read!

The methods were very clear both with regard to procedure and hypotheses. Unfortunately, I didn't find the description in section 2.1 as clear. I think information about what dyad pairs are the target of analysis could come earlier, together with hypothesis. I also first found (both here and in abstract) the information that there are 3 generations but a G4 participant somewhat confusing. Consider revisiting this section.

We have updated the abstract to make clear that the G4 Observer was a confederate early on. Because we have updated the order of the manuscript, the methods now appear before the results, with the design of the study presented in more depth. Section 2.1 is therefore now covered in section 2.5 'Design and Procedure'. Hopefully this new structure makes both design and the analysis plan clearer.

Please clarify in Figure that G4 is confederate.

We have added text to the caption of Figure 4 to clarify that G4 is a confederate.

In Figure 3, please add "psychological" to synchrony label on the x-axis

We have updated Figure 3 (now Figure 4) to reflect this change.

Maybe I missed it, but for psychological synchrony (correlation between expectancy ratings) what were the actual values here? How good were observers at predicting how much pain demonstrators thought they would be in?

We have added the following text highlighting the average strength of association between Observer and Demonstrator expectancies (please see Lines 503-505):

The average standardised coefficient representing the strength of the association between Observer and Demonstrator expectancies had a medium effect size ($M=0.34$, $SD=0.45$), suggesting generally high levels of psychological synchrony between participants.

For psychological synchrony, why is the demonstrator's pain expectancy rating used and not their actual pain? Unless I missed it the argument isn't spelled out in the paper and it would be useful to have it there.

Thank you for raising this point. We have added the following text (please see lines 339-342):

As previous research has demonstrated that expectancies feed into the perception of pain at the level of each trial⁶⁶, we chose to focus on Demonstrator expectancies, rather than pain ratings, as we wanted to measure synchrony for the same cognitive process – belief about the upcoming thermal stimulus – occurring at the same point in time for both participants.

I don't think it is necessary to present the step-wise regression, and variable selection by step-wise methods are generally not recommended. Instead, just present the full model (synchrony, generation, sync X generation), for each of the three synchrony measures.

We agree that stepwise regression may not be recommended for variable selection. However, this was not our intention here. We believe that the presentation of the lower-order coefficients, independent of the interaction, is useful for interpretation when the interaction term is not significant, given that these lower order coefficients can be interpreted as main effects (synchrony across all levels of generation and vice versa), rather than conditional effects as is the case when the interaction is included in the model. Our preference is therefore to present the full models, so readers can appraise each stage should they wish.

Lines 248-249 - is it possible to explain in greater detail why the analyses concern the expression of empathic pain? Additionally, why this is significant to explore.

We have added the following text to clarify the focus on empathetic pain (please see lines 578-582):

It is important to establish a complete understanding of the processes involved during the social transmission of pain, including from the perspective of both Demonstrators and Observers. Direct nociceptive pain experienced by the Demonstrators has been the focus of analysis regarding intrapersonal pain. However, little is known about the vicarious or 'empathetic pain' elicited in the Observers when watching the experience of pain⁷⁶. In achieve this, an exploratory analysis was run concerning the Observer response in primary and secondary outcomes while watching the Demonstrator (see Figure 5).

For Figure 5, please add a figure legend for the meaning of the filled and dashed lines (non-significant and significant?).

We have updated the Figure 6 caption to also refer to the dashed lines (in addition to the filled ones):

Figure 6: Summary of the interpersonal outcomes between dyads, and intrapersonal outcomes at the level of the Observer, that went on to predict that magnitude of that Observer's nocebo effect in the subsequent block

(i.e., when they Demonstrated). Please note that this Figure is for illustrative purposes with each regression path calculated separately (as presented in the analyses above) and not estimated as an overall path model. Filled boxes and arrows represent significant predictors of the subsequent Demonstrator's nocebo effect, while dashed arrows represent non-significant predictors. 'G2 Nocebo' and 'G3 Nocebo' boxes represent this nocebo effect (the difference between Tx and NT Cues). This difference between Cues is denoted as a 'Pain Difference' at G1, due to the genuine difference in nociceptive stimuli. Recursive arrows between 'G2 Nocebo' and 'Psychsynch' boxes represent the significant interaction term between these factors when predicting the subsequent nocebo effect in the chain. Abbreviations are as follows: Psychsynch = Psychological Synchrony; DETsynch = Physiological Synchrony (Determinism CRQA metric); Exp = Expectancy Rating.

Thank you for considering the present manuscript. We look forward to your response.

Sincerely, Dr. Kirsten Barnes

Kirsten Barnes, PhD (kirsten.barnes@unsw.edu.au)

References:

- Barnes, K., McNair, N. A., Harris, J. A., Sharpe, L., & Colagiuri, B. (2021). In anticipation of pain: expectancy modulates corticospinal excitability, autonomic response, and pain perception. *Pain*, 162(8).
https://journals.lww.com/pain/Fulltext/2021/08000/In_anticipation_of_pain__expectancy_modulates_12.aspx
- Greffrath, W., Baumgärtner, U., & Treede, R.-D. (2007). Peripheral and central components of habituation of heat pain perception and evoked potentials in humans. *Pain*, 132(3), 301-311.
<https://doi.org/https://doi.org/10.1016/j.pain.2007.04.026>

20th Nov 23

Dear Dr Barnes,

Your manuscript titled "Your pain increases my pain: Exploring the effect of interpersonal synchrony on the transmission of nocebo hyperalgesia" has now been seen by our reviewers, whose comments appear below. In light of their advice I am delighted to say that we are happy, in principle, to publish a suitably revised version in Communications Psychology under the open access CC BY license (Creative Commons Attribution v4.0 International License).

We therefore invite you to revise your paper one last time to address the remaining concerns of our reviewers and a list of editorial requests. At the same time we ask that you edit your manuscript to comply with our format requirements and to maximise the accessibility and therefore the impact of your work.

Please note that it may still be possible for your paper to be published before the end of 2023, but in order to do this we will need you to address these points as quickly as possible so that we can move forward with your paper.

EDITORIAL REQUESTS:

SUBMISSION INFORMATION:

OPEN ACCESS:

Communications Psychology is a fully open access journal. Articles are made freely accessible on publication under a [CC BY](http://creativecommons.org/licenses/by/4.0) license (Creative Commons Attribution 4.0 International License). This license allows maximum dissemination and re-use of open access materials and is preferred by many research funding bodies.

For further information about article processing charges, open access funding, and advice and support from Nature Research, please visit <https://www.nature.com/commspsychol/article-processing-charges>

At acceptance, you will be provided with instructions for completing this CC BY license on behalf of all authors. This grants us the necessary permissions to publish your paper. Additionally, you will be asked to declare that all required third party permissions have been obtained, and to provide billing information in order to pay the article-processing charge (APC).

* TRANSPARENT PEER REVIEW: Communications Psychology uses a transparent peer review system. On author request, confidential information and data can be removed from the published reviewer reports and rebuttal letters prior to publication. If you are concerned about the release of confidential data, please let us know specifically what information you would like to have removed. Please note that we cannot incorporate redactions for any other reasons.

* CODE AVAILABILITY: All Communications Psychology manuscripts must include a section titled "Code Availability" at the end of the methods section. We require that the custom analysis code supporting your conclusions is made available in a publicly accessible repository at this stage; please choose a repository that generates a digital object identifier (DOI) for the code; the link to the repository and the DOI must be included in the Code Availability statement. Publication as Supplementary Information will not suffice.

* DATA AVAILABILITY:

[link redacted]

Best regards,

Jennifer Bellingtier

Jennifer Bellingtier, PhD
Senior Editor
Communications Psychology

and on behalf of

Yafeng Pan, PhD
Editorial Board Member
Communications Psychology
orcid.org/0000-0002-5633-8313

REVIEWERS' EXPERTISE:

Reviewer #1 pain/empathy for pain
Reviewer #3 social learning/synchrony

REVIEWERS' COMMENTS:

Reviewer #1 (Remarks to the Author):

I want to congratulate the authors to a successful revision! I have no comments left and happily recommend publication in Communications Psychology :)

Reviewer #3 (Remarks to the Author):

I thank the authors for their careful revision of their paper and their responses to my comments on the initial submission. I am happy to recommend the paper for publication and congratulate the authors on this contribution to the literature.